# EFFICIENT PACKING: TOWARDS 2X NLP SPEED-UP WITHOUT LOSS OF ACCURACY FOR BERT

## ABSTRACT

We find that at sequence length 512 padding tokens represent in excess of 50% of the Wikipedia dataset used for pretraining BERT (Bidirectional Encoder Representations from Transformers). Therefore by removing all padding, we achieve a 2x speed-up in terms of sequences/sec. To exploit this characteristic of the dataset, we develop and contrast two packing algorithms. Both algorithms rely on the assumption that sequences are interchangeable and therefore packing can be performed on the histogram of sequence lengths, rather than per sample. This transformation of the problem leads to algorithms which are fast and have linear complexity in dataset size. The shortest-pack-first histogram-packing (SPFHP) algorithm determines the packing order for the Wikipedia dataset of over 16M sequences in 0.03 seconds. The non-negative least-squares histogram-packing (NNLSHP) algorithm converges in 28.4 seconds but produces solutions which are more depth efficient, managing to get near optimal packing by combining a maximum of 3 sequences in one sample. Using the dataset with multiple sequences per sample requires adjusting the model and the hyperparameters to keep the predictive quality of the model. We demonstrate that these changes are straightforward to implement and have relatively little impact on the achievable performance gain on modern hardware. Finally, we pretrain BERT-Large using the packed dataset, demonstrating no loss of convergence and the desired 2x speed-up.

## 1 INTRODUCTION

Since its introduction in 2019, BERT (Devlin et al., 2019a) has been the backbone driving the most exciting advances in Natural Language Processing (NLP). Pre-training BERT from scratch requires substantial computational resources which may be out of reach for researchers and industry professionals. To some extent this has been addressed by the public release of pre-trained models of different sizes and depths (Turc et al., 2019). The introduction of ALBERT (Lan et al., 2019) and Switch transformers (Fedus et al., 2021) further improved the accessibility of larger models. However, the dependence on pre-trained models limits the ability of researchers to explore new backbone architectures. Furthermore, it limits the extent to which practitioners in industry can leverage internal datasets and adapt the model to their particular needs. Hence, any approach that speeds up the pre-training process is desirable from an economical as well as environmental perspective.

In this paper, we present efficient methods to enable researchers to accelerate the pre-training of BERT by as much as 2x without loss of accuracy. The de-facto pre-training dataset Wikipedia, as well as many other NLP datasets, have a skewed distribution of sequence lengths. We show that padding tokens (wasted compute) represent 50% of all tokens of the Wikipedia pre-training dataset at sequence length 512. Thus, by avoiding processing the padding tokens one can get a 2x speed-up. Overall, the lengths range between 5 tokens up to 512 (see Figure 1). Samples of length 512 represent only 23.5% of the dataset, a surprising result given that the pre-processing in BERT attempts to "pack" together sentences so as to fill the sequence length as completely as possible (Devlin et al., 2019c). While processing the padding tokens wastes compute, it is still the most standard approach for leveraging modern massively-parallel compute especially on GPUs. These are most efficient when applying the same operation to each sequence in a batch. By padding all sequences to the same maximum sequence length, they can easily be batched. We note that this naive batching is the most widely used and provided in the vanilla BERT implementation as well as the Hugging Face framework (Wolf et al., 2020) and thus considered as our baseline for comparison.

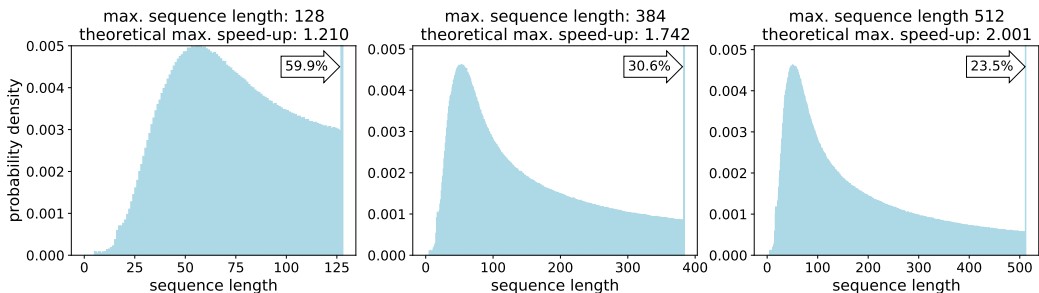

Figure 1: Wikipedia BERT pre-training dataset sequence length histograms (token count excluding padding) for different maximum sequence lengths. Based on the Wikipedia article dump from October 1st 2020. The theoretical speed-up relates to not using any padding tokens and not having any overhead from processing the different lengths.

The most obvious way to reduce the extent of padding in the dataset is to group samples by size before batching (SORT), i.e., process the shorter samples together and longer samples together. BERT is pre-trained in two phases, where the first phase uses sequence length 128 for 900K steps and the second phase uses sequence length 512 for 100K steps. However even by splitting the training in this way, the wasted compute due to padding is approximately 20% (see Figure 1). Other examples of this "sorted batching" approach can be found in Faster Transformer (NVIDIA, 2021), lingvo (Shen et al., 2019) fairseq (Ott et al., 2019), and RoBERTa (Liu et al., 2019), which group samples of similar size together in one batch and fill up with padding only to the maximum length in this batch. This approach can be highly efficient in cases where the dataset length is multiple orders of magnitude larger than the batch size and the number of different sequence lengths. Despite its high computational efficiency, this approach has multiple drawbacks. We outline these below and propose an alternative which maintains the high efficiency, while also circumventing the downsides. Firstly, sorting the data can reduce the overall convergence speed when the batch size is large because it violates the i.i.d. assumption on the data distribution (Bottou et al., 2018; Meng et al., 2019). Secondly, processing batches with shorter sequence lengths under-utilizes the compute compared to running the same batch size with a longer sequence length. For GPUs, a common heuristic to mitigate this effect is to adjust the batch size to keep the number of processed tokens near constant (Ott et al., 2019; Liu et al., 2019). In general however, the relationship between the sequence length and the optimum batch size is more complex and maximizing compute utilization can require the model to be sharded differently across multiple accelerators. Avoiding this, often manual process, is important for ease of use and the portability of methods across different hardware architectures. Thirdly, modern NLP applications are optimized and compiled for fixed tensor sizes using tools such as XLA (XLA, 2021; Fedus et al., 2021), which provides a $\approx 7x$ acceleration for BERT in MLPerf™ (Mattson et al., 2020) compared to the non-XLA baseline (XLA, 2021). Changing the sequence length or batch size requires re-optimization of the computational graph and recompilation of the program for the new tensor shapes. For complex models such as BERT, optimization and recompilation take a non-negligible amount of time. Even if one pre-compiled and cached all combinations of batch size and sequence length, the kernels would still need to be re-uploaded to the device every time the shapes change. Depending on how frequently the tensor shapes change, the overhead from switching kernels adds up. To avoid these issues, it is preferable (and common) to work with fixed tensor shapes for the entire duration of the training run.

More advanced approaches for reducing the padding overhead rely on custom computational kernels. Loosely these are referred to as **"un-padding"** approaches. In Effective Transformer (ByteDance Inc., 2021), the input batch is provided as a padded matrix but padding values are dynamically removed and restored during different calculation stages. While un-padding implementations are highly sophisticated and are able to completely circumvent the processing of padding tokens, they introduce a significant overhead due to the multiple GPU kernel launches (i.e., one kernel per sequence rather than one kernel per batch). Additionally the time to process each batch will fluctuate depending on the sequence lengths in each batch, i.e., batches with only shorter sequences will typically be processed faster. When working with more than one accelerator, this variability in throughput results in all devices in the cluster waiting for the device with the most compute inten-

sive batch to finish processing. As such, un-padding approaches are not appropriate for deployment on large clusters. The **"packing"** based approach introduced in this paper offers significant advantages over un-padding approaches. Firstly, packing is implemented directly at the framework level and requires no additional custom kernel implementations. Secondly, the processing time for each batch is independent of the content of the batch, allowing the packing based approach to maintain the same speed-up whether running on a single device or thousands.

While we demonstrate the effectiveness of packing specifically on the Wikipedia dataset, packing SQuAD (Rajpurkar et al., 2016) or GLUE datasets (Warstadt et al., 2018; Wang et al., 2018) for BERT also leads to significant speed-ups (some in excess of 9x) (Sections H and I). The effectiveness of packing is a result of both the length distribution of the documents in the source datasets as well as the different text preprocessing steps for BERT (Devlin et al., 2019c). The use of bi-directional self-attention in BERT implies that the input sequences should contain complete sentences. If a sentence is abruptly cut short, the hidden state on other (preceding) tokens in the sequence will be affected. Language models with causal attention (only attending to previous tokens in the input) do not have this issue to the same degree. For such models, if a sequence is cut short at an arbitrary token, the other tokens (which occur earlier in the sequence) will not be affected. This ability to cut sequences arbitrarily completely trivializes the packing problem for models based on causal attention. For instance, GPT-3 (Brown et al., 2020) is trained with a maximum sequence length of 2048 where a single sequence may contain multiple segments of sentences separated by a special end of segment token. The last segment in each sequence is simply cut to meet the sequence length requirement making the packing problem trivial and avoiding any padding. In the interest of computational efficiency GPT-3 does not mask the attention between different segments in a sequence. In contrast, the packing approach presented in this paper introduces a mask in the attention layer (see Section 3.2.2) to prevent cross-contamination between examples in a pack. Note, we mask the interaction between different sequences and not between different sentences or segments in the same sequence. This ensures that the characteristics of the original dataset and model are matched as closely as possible. RoBERTa and many other models in production like T5 (Raffel et al., 2019) use a similar packing approach as GPT-3, combining full sentences/sequences with GREEDY packing (first come first concatenate) and also separation tokens or additional padding. The RoBERTa ablation study shows that mixing of sentences from different documents reduces accuracy, but it is used nonetheless for load balancing reasons which indicates that sorted batching is not sufficient.

In summary, the contributions of the paper are as follows. In Section 2, we produce histograms of the Wikipedia pre-training dataset showing the high percentage of padding tokens. We present two new deterministic and efficient packing algorithms which efficiently pack datasets with millions of sequences in a matter of seconds (or less) in Section 3.1. We empirically show that the proposed packing algorithms produce a nearly-optimal packing scheme In Section 3.2 and Section 3.3, we explain how the BERT model can be adjusted to show the same convergence behavior on packed and unpacked sequences. In Section 4.2, we demonstrate that the convergence of the BERT large model on the packed dataset is equivalent to that on the un-packed dataset with 2x throughput increase on the Wikipedia sequence length 512 pre-training dataset.

## 2 WIKIPEDIA BERT PRE-TRAINING DATASET

BERT is pre-trained using masked-language modelling and next-sentence prediction on a large corpus of Wikipedia articles. Each sequence is composed of one <CLS> token followed by the first "segment" of sentences, followed by a <SEP> token, and then finally the second "segment" of sentences. Because these "segments" are created in sentence-level increments there is no token-level control of sequence length. Furthermore $10\%$ (default value, (Devlin et al., 2019b)) of sequences are intentionally cut short. This leads to significant levels of padding, especially for longer maximum sequence lengths (see Figure 1). At sequence length $128$ (commonly used in phase 1 of pre-training) the theoretical speed-up is around $1.2$, at sequence length $384$ this increases to $1.7$, and finally at sequence length $512$ (commonly used for phase 2 of pre-training) it is $2.0$. Despite the widespread use of the Wikipedia dataset for pre-training BERT such histograms have, to the best of our knowledge, not been published previously. This has perhaps lead to the underestimation of the speed-up opportunity available. To put things into perspective, the sequence length 512 dataset contains 8.33 billion tokens, of which 4.17 billion are padding tokens.

## 3 METHODS

Our approach consists of three distinct components. Firstly, we pack the $n$ data samples efficiently during pre-processing to make full use of the maximum sequence length, $s_m$ (Sections 3.1.1, 3.1.2, and E). Secondly, we introduce a series of model changes in Section 3.2 that preserve the equivalence with the original BERT implementation. The changes include a self-attention mask to prevent the model from attending between different sequences in the same pack (Section 3.2.2), and an adjustment of the the positional embeddings (Section 3.2.1) to handle packs of sequences. Other components of the model, such as the feed-forward layer (Vaswani et al., 2017), operate on a per-token basis and do not require modification for pre-training. In Section 3.2.3, we also demonstrate how to compute a per-sequence loss and accuracy for NSP and downstream fine-tuning tasks. Thirdly, we provide suggestions for hyperparameter adjustment (Section 3.3) that lead to analogous convergence behavior between the packed and un-packed BERT implementations.

### 3.1 PACKING ALGORITHMS

The problem of optimally concatenating multiple sequences of different length until a maximum combined length is reached can be directly framed as a bin-packing problem. Since an exact solution is strongly NP-complete (Korte & Vygen, 2012), we propose two new heuristic algorithms that are tailored to the NLP setting. A detailed introduction to packing is provided in Section E.

#### 3.1.1 SHORTEST-PACK-FIRST HISTOGRAM-PACKING (SPFHP)

Shortest-pack-first histogram-packing (SPFHP) works on the bins in the sequence length histogram (with bin size 1) rather than the individual samples. The histogram is traversed in sorted order from longest to shortest sequences. Then, to pack the data during the traversal, we apply the worst-fit algorithm (Johnson, 1973; Yue & Zhang, 1995) such that the histogram bin being processed goes to the **"pack"**[1] that has the most space remaining ("shortest-pack-first"). If the histogram bin does not fit completely, a new pack is created. We also limit the **packing depth**, in other words the maximum number of sequences that are allowed in a pack. Therefore, an existing pack is only extended if it is not already at maximum packing depth. The detailed code for the algorithm is provided in Listing 3. The time and space complexity of the algorithm are $O(n + s_m^2)$ and $O(s_m^2)$ (Section F.2).

#### 3.1.2 NON-NEGATIVE LEAST SQUARES HISTOGRAM-PACKING (NNLSHP)

The proposed NNLSHP algorithm is based on re-stating the packing problem as a (weighted) non-negative least squares problem (NNLS) (Bro & De Jong, 1997) of the form $wAx = wb$ where $x \geq 0$. The vector $b$ is the histogram containing the counts of all the sequence lengths in the dataset. Next, we define the $A$ matrix (the "packing matrix") by first generating a list of all possible sequence length combinations ("strategies") that add up exactly to the maximum sequence length. We focus specifically on strategies that consist of at most 3 sequences per pack (independent of $b$) and encode each strategy as a column of the sparse matrix $A$. For example, a strategy consisting of the sequence length 128, 128, and 256 in represented a column vector that has the value 2 at the 128th row, the value 1 at the 256th row, and zero at all other rows. The variable $x$ describes the *non-negative* repetition count for each strategy. So a 24 in the $i$th row of $x$ means that the strategy represented by the $i$th column of $A$ should repeat 24 times. Moreover, in the un-weighted setting, $Ax = b$ states that we would like to "mix" the pre-defined strategies (columns of $A$) such that the number of samples matches the histogram $b$, and where each strategy is used $x \geq 0$ times. We use the residual weight $w$ to control the penalization of the $Ax - b$ residual on different sequence lengths (different rows of $b$). Heuristically, we set the weight of $0.09$ for all sequences of length 8 or smaller because they are considered acceptable padding sequences while all other sequence lengths get weight 1. We discuss this heuristic choice of parameters in Section E.4.5 and E.5. The overall efficiency of the packing is not greatly influenced by the weighing (less than $1\%$ extra speed-up).

After solving $wAx = wb$ for $x \geq 0$ using an off-the-shelf solver, we obtain a floating point solution, which means that the repetition counts are not necessarily integers. Since we cannot use a non-natural number of strategies, we round the solution $\hat{x}$ to the nearest integer. The error introduced

---

[1]We avoid the ambiguous terms "bin" and "sample/sequence"and use "pack" instead to refer to the multiple sequences concatenated during packing.

by this rounding is found to be negligible (a few hundred sequences in the worst case) compared to the size of the dataset (millions of sequences). The time complexity and space complexity of the algorithm are $O(n + s_m^5)$ and $O(s_m^3)$. Further details are provided in Section E.4.

## 3.2 PACKEDBERT: MODEL CHANGES

This section describes how any vanilla BERT implementation should be modified for packed sequence processing, such that the behavior of the model is the same as when processing unpacked sequences. Preserving the mathematical equivalence is necessary to ensure existing BERT pre-training and fine-tuning practices remain valid, as well as being required by benchmarks such as MLPerf™ (Mattson et al., 2020).

### 3.2.1 POSITIONAL EMBEDDINGS FOR PACKED SEQUENCES

The BERT model uses three types of embeddings: token, segment, and positional embeddings. The latter is canonically implemented as a bias add operation, rather than a full embedding look-up. This is possible because the positional indices are the same for every sequence. However, when using the packed data format the position index needs to be reset with each new packed sequence. For instance, when packing two sequences one of length 2 and one of length 3, the positional embedding indexes that need to be picked up are $[0, 1, 0, 1, 2]$. To achieve this, the bias add needs to be replaced by an embedding look-up to extract the correct positional embedding for each token in the pack. This also requires keeping an extra input which specifies the position of each token in its sequence. This adjustment has only a minor impact on absolute accuracy/loss but is required to reach the target accuracy (Section 4.2 and C).

### 3.2.2 ATTENTION MASKING FOR PACKED SEQUENCES

To maintain an implementation that is consistent with the un-packed version, tokens from different sequences within a pack should not be able to attend to each other. This is typically achieved in other implementations by unpacking the sequences using custom attention kernels and then doing the attention per-sequence (ByteDance Inc., 2021). Instead, we propose directly masking the attention matrix with a block-diagonal mask before the attention softmax. This is straightforward to implement in modern frameworks (see Figure 2). Naturally, there is a cost to both the mask construction and applying it to the attention matrix (see Table 1, Section 4.1). However, it is required to keep the accuracy (Section 4.2 and C).

```
1  mask = np.array([[1, 1, 1, 2, 2]])  # input
2  zero_one_mask = tf.equal(mask, mask.T)  # 0, 1 mask
3  # for use with softmax:
4  softmax_mask = tf.where(zero_one_mask, 0, -1000)
```

$$\begin{pmatrix} 1 & 1 & 1 & 0 & 0 \\ 1 & 1 & 1 & 0 & 0 \\ 1 & 1 & 1 & 0 & 0 \\ 0 & 0 & 0 & 1 & 1 \\ 0 & 0 & 0 & 1 & 1 \end{pmatrix}$$

Figure 2: Attention mask code sample [left] and example zero-one mask [right].

### 3.2.3 CALCULATING PER-SEQUENCE LOSS AND ACCURACY

Canonical implementations of BERT compute the cross-entropy loss for the masked language model on a per-token basis. However other NLP tasks, such as SQuAD, compute the loss and accuracy on a per-sequence basis. This section discusses how to handle such tasks when training with packed sequences. Simply feeding packs of sequences to the same implementation of cross-entropy would result in a per-pack weighted loss. In other words, the overall loss on the micro-batch would sum-up the losses on the individual packs, rather than individual sequences. As a result, the model would converge to a different optimum than when running with the un-packed implementation. For instance, a pack of a single sequence would contribute to the loss with the same weight as a pack of three sequences.

To recover the per-sequence averaging behavior of the canonical un-packed BERT implementation, we effectively "unpack" the incoming logits and labels. Once the sequences have been unpacked, we can compute the loss on each sequence separately as usual and then add up the losses. However, rather than looping through the sequences index, we compute on all indexes in parallel (see

Figure 3). This minimizes the latency overhead of un-packing the loss calculation. As an example, we show how per-sequence loss can be implemented for the pre-training task. We use the "masked lm weight" (Devlin et al., 2019b) input tensor to represent which sequence a given masked token belongs to (0, 1, 2 and so on). This is consistent with the canonical BERT implementation where this input takes a value of either 1 (belonging to the sequence) or 0 (belonging to padding). The full methodology is detailed in Listing 5 and can be applied to other classification or pre-training tasks.

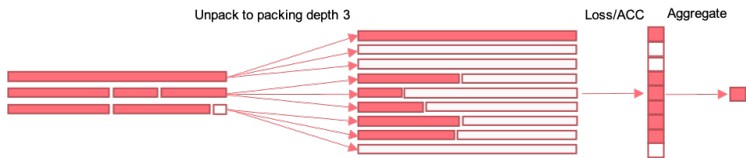

Figure 3: Vectorized unpacking of the sequence loss. White rectangles correspond to padding.

### 3.3 HYPERPARAMETER ADJUSTMENT

In terms of convergence behavior, the primary consequence of packing is an increase in the effective batch size (with respect to number of sequences and real tokens) with some variation over different iterations. When the loss is averaged per-sequence, if each pack on average contains two sequences, the batch size (per optimization step) is effectively doubled on average. Similarly, for per-token averaged losses depends on the effectiveness of the compression. Reducing the fraction of padding tokens in the dataset from 50% to 0% results in there being twice as many tokens in the batch i.e. the dataset is compressed by a 2x factor. While one could subsequently reduce the computational batch size by the packing factor (average number of sequences per pack) and keep using the same hyperparameters, this is typically not desirable as it might imply under-utilizing the memory/compute.

Instead, we propose an approximate heuristic for updating the decay parameters of the LAMB optimizer (You et al., 2019). For a packed dataset with a packing factor $p$, we update the decay parameters as: $\beta_1 := \beta_1^p$, $\beta_2 := \beta_2^p$. For $p = 2$, this corresponds to the exact parameters for calculating momentum and velocity, when updating with the same gradient twice (Section A). A common approach is to scale the learning rate with the batch size. However, our experiments in Section 4.2 show that this reduces convergence speed.

Since these adjustments are only heuristics the convergence of the model will be comparable but not identical. In particular, it is unlikely that simply adjusting the hyperparameters will fully undo the impact of the increased batch size. However, with these adjustments, researchers should be able to continue to use existing configurations.

## 4 EXPERIMENTS

### 4.1 BIN-PACKING ALGORITHM COMPARISON

We evaluate our algorithms using the following metrics: **number of packs**, **number of all tokens**, **number of padding tokens**, **solution time of the packing algorithm** (after histogram and strategy creation), **number of strategies used**, **packing efficiency** (the fraction of non-padding tokens in the packed dataset), the **speed-up** achieved compared to not packing (depth 1), and the average number of sequences per sample (**packing factor**). For SPFHP, we analyse different (maximum) packing depth, since packing is less efficient with smaller depth and we want to get a general understanding on how the packing depth influences the processing time. For NNLSHP, we focus on packing depth 3 because it packs the data sufficiently well.

For the speed-up analysis, we focus on the intelligence processing unit (IPU) (Jia et al., 2019) (IPU-M2000, 16 accelerator chips). A GPU dynamically loads the code into the accelerator; in contrast, the IPU works with a static pre-compiled engine that gets loaded onto the chip at the start of the run. While other approaches result in excessive padding or continuous changes of the code, our approach can work with the same code for the whole dataset. So in this setting the IPU architecture would

especially benefit from our approach since it avoids code changes. Nevertheless, it can be applied to any implementation on GPU or TPU. For determining the speed-up, we take advantage of the precompiled kernel. Since time measurements are quite noisy, we can profile the kernel and how many cycles it takes for processing a batch. That way, we can determine the **overhead** (in cycles) from processing the additional attention masking and for unpacking the loss. Combining **overhead** and **packing factor**, we get the **speed-up** estimate. No experiment repetitions are required since the algorithms and measurements are deterministic.

The main results for the performance metric evaluation are displayed in Table 1. The processing time for SBFHP was around $0.03s$ and independent from the packing depth. We see that the overhead slightly increases with packing depth but that the benefits of packing outweigh the cost. The best speed-up is obtained with NNLSHP at depth 3 which required $28.4s$ for processing and ran out of memory for larger depth. With a value of $1.913$, it is close to the theoretical upper bound of $2.001$. The results show that efficiency, packing factor, and speed-up can be viewed inter-changeably. The amount of time needed to process a sample (a pack of sequences) is barely changed relative to the un-packed implementation. The packing factor or the improvement in efficiency effectively provide an accurate estimate of the speed-up.

Table 1: Key performance results of proposed packing algorithms (SPFHP and NNLSHP).

| packing depth | packing algorithm | # packs [M] | efficiency (%) | packing factor | overhead (%) | realized speed-up |
|---|---|---|---|---|---|---|
| 1 | NONE | 16.280 | 49.97 | 1.000 | 0.000 | 1.000 |
| 1 | SORT | 16.280 | 99.99 | 2.000 | 100 | 1.000 |
| $\approx 10$ | GREEDY | $\approx 10.397$ | $\approx 78.24$ | $\approx 1.566$ | $\approx 4.48$ | $\approx 1.5$ |
| 2 | SPFHP | 10.102 | 80.52 | 1.612 | 4.283 | 1.544 |
| 3 | SPFHP | 9.095 | 89.44 | 1.790 | 4.287 | 1.716 |
| 3 | NNLSHP | 8.155 | 99.75 | 1.996 | 4.287 | **1.913** |
| 4 | SPFHP | 8.659 | 93.94 | 1.880 | 4.294 | 1.803 |
| 8 | SPFHP | 8.225 | 98.90 | 1.979 | 4.481 | 1.895 |
| 16/max | SPFHP | 8.168 | 99.60 | 1.993 | 4.477 | 1.905 |

**Packing depth** describes the maximum number of packed sequences. NONE is the baseline BERT implementation, whereas SORT corresponds to sorted batching, and GREEDY concatenates sequences as they arrive until they exceed $512$ tokens. Setting no limit resulted in a maximum packing depth of 16. The **number of packs** describes the length of the new packed dataset. **Efficiency** is the percentage of real tokens in the packed dataset. The **packing factor** describes the resulting potential speed-up compared to packing depth 1. With **overhead**, we denote the percentage decrease in throughput due to changes to the model to enable packing (such as the masking scheme introduced in Section 3.2.2). The **realized speed-up** is the combination of the speed-up due to packing (the **packing factor**) and the decrease in throughput due to the **overhead**. It is used to measure the relative speed-up in throughput and the overhead from masking and loss adjustment.

## 4.2 LEARNING CURVES AND HYPERPARAMETER ADJUSTMENT

For depth 1 (classic BERT) and NNLSHP with depth 3, we additionally evaluate on the MLPerf™ version 0.7 BERT pre-training benchmark (Mattson et al., 2020). Briefly, this involves training from a standard checkpoint to a masked-language model accuracy of $71.2\%$ using 3 million sequences with a maximum length of $512$ tokens (refer to MLCommons (2020) for details). Following this standardized benchmark supports reproduction of results even on other systems and makes sure that the reproduction effort is moderate and setup rules are clearly documented. We compare the resulting speed-up as well as the respective learning curves by evaluating the data on a held-out validation dataset. The objective of this additional evaluation is to analyse if convergence behavior is changed by the packing strategy and if the theoretical speed-up can be achieved in practice.

With packing, we effectively increase the average batch size by the packing factor ($\approx 2$). However, with a different batch size, different hyperparameters are required (see Section 3.3) and there is no mapping that will generate exact matching of results but only heuristics. In a first comparison, we use the same hyperparameters when comparing packed and unpacked training except for cutting the accumulation count by half. This way, we make sure that the batch size is constant on **average**.

In the second comparison, we evaluate our heuristics and how they compensate the difference in batch size. This setup is more desirable because it is beneficial to use the hardware to its full potential and cutting the batch size by half usually reduces throughput. In the third comparison, we compare two optimized setups.

The learning curves are displayed in Figure 4. In the first setup, we see the curves almost matching perfectly when normalizing by the numbers of samples processed. Differences can be explained by the variation of the number of sequences in the packing batch, and general noise in the training process. Especially after the initial phase, the curves show a near-identical match. The second setup shows bigger differences since changing the batch size and hyperparameters changes the training dynamics. We observe slower convergence early on in training due to the increased batch size. This is expected. The adjustment of the learning rate actually decreases performance probably because we correct for the increased number of sequences already in the modified loss. With the adjustment of the decay parameter of LAMB, we see matching performance at the later training stages. However, it is not feasible to completely recover the early convergence behavior of the smaller batch size by adjusting the hyperparameters. For instance doubling the batch size of unpacked BERT to 3000 and adjusting the LAMB decay parameters leads to more of a slow down in convergence than when running packed BERT with a batch size of 1500 and a packing factor of 2. n practice, our implementations exceeds the estimated 1.913 maximum speed-up. This estimate is based on the reduction in the computational work needed to process the dataset. However, packing the data also reduces the latency of the transferring the data to the device. Figure 4 shows that the realized total speed-up from packing exceeds 2x. On Squad 1.1 after full packed pretraining, F1 score is reduced by $0.003\%$ whereas the EM score is increased by $0.049\%$ (Section D).

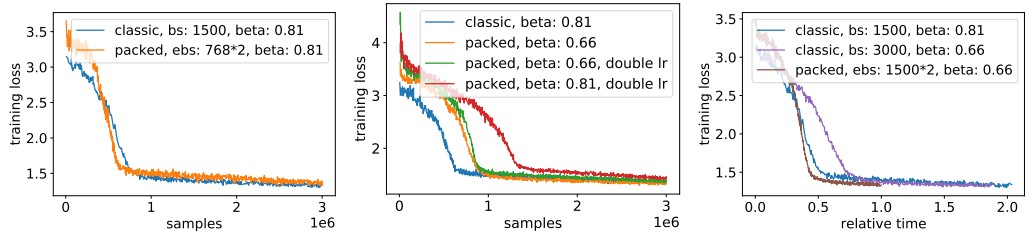

Figure 4: Comparison of learning curves for packed and unpacked processing, where all experiments converged to the target accuracy within the same number of training samples(3 million). [left] same **e**ffective **b**atch **s**ize (**ebs** is batch size times packing factor), [middle] different heuristic adjustments of the hyperparameters (batch size 1500 for all runs, such that **ebs** for packed runs is $1500 * 2$), and [right] realized speed-up from packing (in excess of desired 2x).

### 4.3 SCALING ANALYSIS: IMPACT OF THE NUMBER OF ACCELERATORS

A further advantage of packing over competing un-padding approaches is the inherent load balancing provided by packing. So called un-padding approaches rely on dynamically launching custom kernels that ignore padding. A stated advantage of such implementations is the ability to avoid computing the complete (512 x 512) attention matrix. This provides additional computational savings compared to packing, where the attention matrix is computed in its entirety and then masked. Because of these additional savings, un-padding can exceed the theoretical upper bound for speed-up from packing (2.013 on Wikipedia). As a result of the dynamic nature of the approach, the processing time with un-padding is different for each sequence in the batch, and the amount of time required to process a batch of sequences will be determined by the processing time of the longest sequence in the batch (with the sequences being processed in parallel). Furthermore, in the multiple accelerator setting the processing time on each device will vary depending on the sequences in the batch that it receives. Devices which finish early have to wait for the slowest device to finish before exchanging gradients. This load-imbalance between the devices (and inside the batch) leads to a considerable decrease in the speed-up from un-padding as the number of accelerators is increased (see Figure 5).

In contrast, packing (our approach) is inherently load-balanced. The processing time on each accelerator is independent of the content inside the batch received by the device. Any number of accelerators can therefore operate in unison without having to wait for the slowest batch to process

(all per-device batches are equally fast). To demonstrate the severity of the load-imbalance issue, we consider the scaling of an un-padding approach with a per-device batch size of 32 running on eight devices (NVIDIA, 2020). From there, we readily extrapolate the performance to both larger and smaller cluster sizes by fitting a Gumbel distribution to the observed processing times (Section B). On a single device with batch size 32 un-padding outperforms packing and exceeds the theoretical upper-bound for packing. As the number of devices increases to two or more, the proposed packing approach outperforms the dynamic un-padding approach. On a cluster with 32 accelerators the speed-up from un-padding drops to $50\%$ and with $2048$ devices the speed-up is only $30\%$. In contrast, the speed-up due to packing is independent of the number of accelerators and stays at $1.913$. Switching to a smaller batch size would reduce the load-imbalance issue to some extent, but would also result in under-utilization of the available memory and compute.

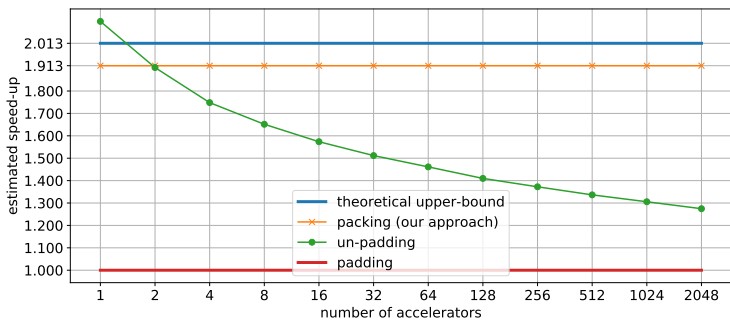

Figure 5: Comparison of the theoretical speed-up achievable as the number of accelerators is increased.

## 5 CONCLUSION

We showed that packing can be easily implemented without the need for any custom kernels while still providing a 2x speed-up without a loss of accuracy. Additionally, we showed that any additional speed-ups resulting from dynamic un-padding approaches diminish for even moderate batch sizes or when additional accelerators are added. In contrast, packing is load-balanced and maintains the 2x throughput when scaling to large numbers of accelerators. Furthermore, the computational overhead introduced by the positional embedding, the attention mask, and potentially the packed per-sequence loss are small compared to the achieved acceleration. This overhead remains below $5\%$ on the IPU for all tested packing depths. The efficient packing algorithms presented in this paper enable us to efficiently pack millions of sequences in a matter of seconds. Compared to both the pre-processing time for the Wikipedia dataset and the training runtime, this overhead is negligible. Furthermore, we showed that performing packing as a pre-processing step does not significantly impact the training convergence. Our proposed hyperparameter adjustment scheme additionally helps practitioners easily modify existing validated optimizer settings for use with packed BERT. Further exploration of hyperparameter selection is left to future work.

When performing packing as a pre-processing step, the proposed NNLSHP and SPFHP methods achieve near optimal compression efficiency. In this offline setting, we are able to build a histogram of the dataset, and thus achieve linear time complexity with respect to the number of samples. This makes packing modern datasets with millions of sequences possible. In the future, it would be interesting to extend SPFHP to the online setting where a histogram of the entire dataset cannot be built. Another interesting direction is the packing of images of different sizes to help accelerate computer-vision applications. This is especially relevant given the recent advances in the use of transformer-based approaches in the computer vision domain, for example the visual transformer (Wu et al., 2020). Masking out the self-attention within transformers is easier to implement than avoiding cross-contamination of convolutions applied to packed images. Future work should explore improving the performance of other models (RoBERTa, GPT-3, T5) by avoiding contamination between non-contiguous segments from different documents. Even BERT itself might benefit from avoiding contamination between the two concatenated segments.

## REPRODUCIBILITY STATEMENT

All code for the packing algorithms is available in the appendix (Section O) and is directly linked to our GitHub page to simplify the download and usage. We even provide code for different variants and the histograms of sequence length for different datasets that got tokenized for BERT training of fine-tuning.

To generate the learning curves, our public submission to MLPerf™ could be used and we are preparing further code releases in other frameworks. To encourage the use of the adjustments of models for packed sequences, we additionally provide detailed explanations and code snippets in TensorFlow.

Detailed mathematical formulas (Section B and E), a theorem proof (Section A), and complexity calculations (Section F) are provided in the appendix to support our claims in this paper in full detail.

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

# Appendix

## TABLE OF CONTENTS

BROADER IMPACT

We showed that when pre-training BERT on Wikipedia, the computational overhead taken to process padding tokens is roughly $50\%$. By eliminating this wasted computational time, the approach presented in this paper paves a way to halving the carbon footprint of training BERT-based models.

Furthermore, our approach circumvents the need for custom kernels, making the benefits of packing readily accessible to a broader audience of NLP practitioners. As such, we are hopeful the research will have a positive impact on the NLP community, and do not see any disadvantage of using this approach.

The benefit of our algorithm is based on two assumptions: A skewed length distribution in the training dataset and a hardware setup that trains efficiently on a fixed batch size. If *efficient training* is possible, with a variable batch size approaches like FasterTransformer and the fairseq sorted batch approach will result in the same or even larger benefits (due to smaller self-attention matrices). If the dataset is generated differently like in GPT models (Brown et al., 2020) and RoBERTa (FULL-SENTENCES) (Liu et al., 2019), all sequences will be at full length and sequences cannot be concatenated and there is indeed no benefit in packing sequences. However, strategies that reach full sequence length usually combine segments from different unrelated document sources which can result in reduced performance. Even in the normal BERT model, there might be this contamination between segments from different documents. Our paper introduced an approach to avoid the contamination between sequences. However, the same approach could also be applied to avoid contamination between segments and it remains future work to explore its benefits beyond BERT pretraining.

Future work would need to investigate the applicability of packing on text produced by different cultures and in different languages. We have already shown that the speed-up resulting from using our methods does not only occur when pre-training BERT on Wikipedia but also on other datasets such as SQuAD and GLUE. Furthermore, the sentence length distribution of the original English language text shows similar characteristics. Our research leads us to believe that compressible distributions arise naturally in language tasks and beyond, for instance in DNA sequence lengths (Hansen et al., 2017), protein lengths (Guillén et al., 2013), and speech (Section J). Many such sequence modelling workloads are based on variations of the BERT/transformer architecture and would therefore easily benefit from our acceleration.

Failures in NLP can have a big impact on society; many technologies, such as Alexa, Siri, and Google Home, rely on them. Whilst any errors arising from our approach can be avoided, one potential source of error comes from the implementation. Both the attention mask and the per-sequence loss need to be modified to support packing. These changes are significantly smaller than those required by custom kernels, however they may still be time consuming to implement and debug. To help mitigate the risk of any implementation errors, we share our reference implementations of the required changes in the appendix.

## A   THEOREM ON LAMB HYPERPARAMETER CORRECTION HEURISTIC

With packing, the effective batch size changes and hence hyperparameters of the LAMB optimizer (You et al., 2019) need to be adjusted. For a packed dataset with a packing factor $p$, we update the decay parameters as: $\overline{\beta_1} := \beta_1^p$, $\overline{\beta_2} := \beta_2^p$. For instance if $\beta_1 = 0.81$ for the un-packed dataset, then for a packed dataset with an average of 2 sequences per sample one should use a value of $0.81^2 \approx 0.66$ instead. Assuming no or only minor changes in gradients and $p$ being a natural number, we can prove that this heuristic is the exact solution to make sure that momentum and velocity in LAMB are unaffected by packing. This can be proven by mathematical induction. Note that $p \geq 1$ by definition.

**Theorem 1.** *For any $p \in \mathbb{N}$ and assuming that respective gradients on a batch of $b$ random samples are (approximately) the same, choosing*

$$\overline{\beta_1} := \beta_1^p \tag{1}$$

$$\overline{\beta_2} := \beta_2^p. \tag{2}$$

*as hyperparameters in the LAMB optimizer ensures that the momentum and velocity after $p$ separate update steps are the same as with one packed update step with $p \times b$ samples.*

*Proof.*

- *Base Case*:
  For $p = 1$ the left and right side of the equation are the same which matches exactly the unpacked case. Hence, the theorem holds for $p = 1$.

- *Inductive hypothesis*: Suppose the theorem holds for all values of $p$ up to some $k$, $k \geq 1$.

- *Inductive proposition*: The theorem holds for $p = k + 1$.

- *Proof of the inductive step*: Let $l$ be the loss function, $w_t$ the weight vector after $t$ updates, and $x_1^t, \ldots, x_b^t$ the respective underlying data to calculate the gradient $g_t$. For a single update step in LAMB with batch size $b$ samples, we compute the gradient

$$g_t = \frac{1}{b} \sum_{i=1}^{b} \frac{\partial l}{\partial w}(x_i^t, w^t). \tag{3}$$

Since $g_1 \approx g_2 \approx \ldots \approx g_{k+1}$, We have with the inductive hypothesis and the definitions in LAMB:

$$m_k = \beta_1^k m_0 + (1 - \beta_1^k)g_1 \tag{4}$$

$$v_k = \beta_2^k v_0 + (1 - \beta_2^k)g_1^2 \tag{5}$$

Now we can calculate (with $g_1 \approx g_{k+1}$)

$$m_{k+1} = \beta_1 m_k + (1 - \beta_1)g_{k+1} \tag{6}$$

$$\approx \beta_1 \left( \beta_1^k m_0 + (1 - \beta_1^k)g_1 \right) + (1 - \beta_1)g_1 \tag{7}$$

$$= \beta_1^{k+1} m_0 + (1 - \beta_1^{k+1})g_1 \tag{8}$$

The calculation for $v_k$ is the same. As reference for a packed update with $p = k + 1$ with $\overline{\beta_1}$ and $\overline{\beta_2}$, we would get

$$g = \frac{1}{pb} \sum_{j=1}^{p} \sum_{i=1}^{b} \frac{\partial l}{\partial w}(x_i^j, w^1) = \frac{1}{p} \sum_{j=1}^{p} \left( \frac{1}{b} \sum_{i=1}^{b} \frac{\partial l}{\partial w}(x_i^j, w^1) \right) \approx \frac{1}{p} \sum_{j=1}^{p} g_1 = g_1 \quad (9)$$

since we are calculating gradients over $b$ samples which are assumed to be approximately the same. Consequently, the updates for momentum and velocity would be

$$\overline{m_k} = \overline{\beta_1} m_0 + (1 - \overline{\beta_1})g_1 \tag{10}$$

$$\overline{v_k} = \overline{\beta_2} v_0 + (1 - \overline{\beta_2})g_1^2. \tag{11}$$

Hence, $\overline{\beta_1} = \beta_1^{k+1}$ and $\overline{\beta_2} = \beta_2^{k+1}$ is required to map to the formula with the consecutive updates (for the same amount of data).

- *Conclusion*: The theorem holds for any $p \in \mathbb{N}$.

$\square$

Since we proved that the formulas $\beta_1 := \beta_1^p$, $\beta_2 := \beta_2^p$. hold for all $p \in \mathbb{N}$, $p \geq 1$, it is safe to assume that it is an appropriate heuristic for all $p \in \mathbb{R}$, $p \geq 1$.

## B  UN-PADDING SCALING ESTIMATE

Firstly, we retrieve the per-batch processing time for an un-padding implementation running pre-training on the Wikipedia dataset from (NVIDIA, 2020). These processing times were obtained using 8 GPUs each with a per-device batch size of 32. We also retrieve the throughput numbers for the same system running with padding from (NVIDIA, 2021) and use that as the baseline to compare the un-padded throughput against.

The throughput on the 8 GPU system is effectively limited by the slowest of the eight batches being processed in parallel. The Gumbel distribution is particularly suited to modelling the maximum or minimum value of a fixed size collection of i.i.d. samples (in this case batches). We observe that on 8 GPUs the throughput (i.e. speed-up) distribution indeed closely resembles a Gumbel distribution with $\alpha_1 = 1.6$ and $\beta_8 = 0.13$ as shown in Figure 6.

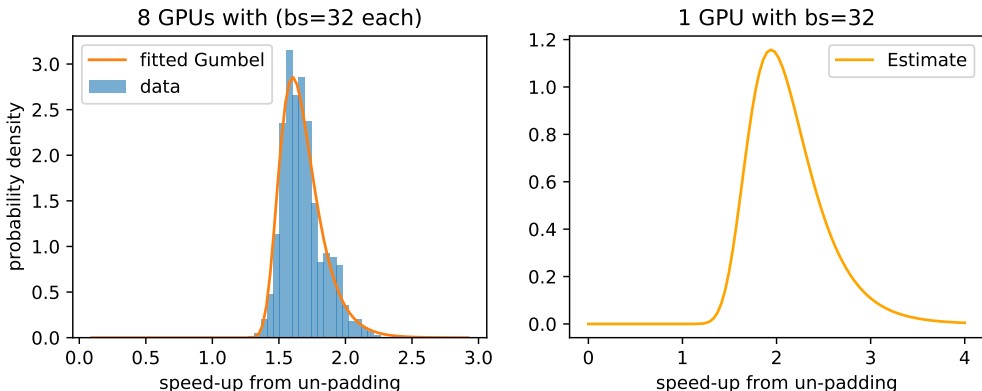

Figure 6: Left: Speed-up from un-padding on 8 GPUs closely resembles a Gumbel distribution. Right: statistical estimate of speed-up distribution on a 1 GPU system running un-padding

We can extrapolate the performance on the 8 GPU system to larger clusters by recognizing that the processing time for each cluster is effectively determined by the slowest batch being processed. Specifically, we could randomly sample (without replacement) two processing times for the 8 GPU system, and record the max of the two as the processing time for a system of 16 GPUs. However, this simple approach is too sensitive to outliers in the data and would result in an under-estimate of the performance of un-padding on large systems. We mitigate the effect of outliers in the data by avoiding directly sampling the processing times. Instead, we fit a Gumbel distribution to the processing times of a single batch of size 32 running on one GPU. To perform the fit, we observe that the cdf on one GPU ($P_1$) is related to the cdf on 8 GPUs ($P_8$) through (Kotz & Nadarajah, 2000)(section 1.3):

$$(1 - P_8(s)) = (1 - P_1(s))^8 \tag{12}$$

In other words, if the speed-up on the cluster is larger than $s$, this implies that the speed-up on every GPUs in the cluster was at least $s$. Assuming $P_1$ is Gumbel and given the 8 GPU Gumbel parameters $\alpha_8$ and $\beta_8$, we need to fit two parameters, $\alpha_1$ and $\beta_1$. Consequently for the median ($s = \alpha_8 - \beta_8 \ln(\ln(2))$, $P_8(s) = 0.5$), we have:

$$0.5 = (1 - P_1(\alpha_8 - \beta_8 \ln(\ln(2))))^8 . \tag{13}$$

And since $P_8$ is Gumbel, we also have an equation for the mode ($s = \alpha_8$, $P_8(s) = e^{-1}$):

$$(1 - e^{-1}) = (1 - P_1(\alpha_8))^8 . \tag{14}$$

We solve these two non-linear equations simultaneously using the standard SciPy optimization package.

Listing 1: Infer Gumble distribution parameters.

```
1  import numpy as np
2  from scipy import stats, optimize
3  alpha_8 = 1.6038
4  beta_8 = 0.1288
5  def g(x):
6      alpha_1, beta_1 = x
7      dist = stats.gumbel_r(loc=alpha_1, scale=beta_1)
8      # Equations for median and mode
9      median = alpha_8 - beta_8*np.log(np.log(2))
10     equation1 = 0.5 - dist.sf(median)**n_gpu
11     mode = alpha_8
12     equation2 = (1-np.exp(-1)) - dist.sf(mode)**n_gpu
13     return (equation1**2 + equation2**2)
14
15 res = optimize.minimize(g, [alpha_8, beta_8], method="Nelder-Mead")
16 alpha_1, beta_1 = res.x
```

The resulting estimated speed-up Gumbel distribution for a single device has $\alpha = 1.94$, $\beta = 0.108$ and is shown in Figure 6 [right]. To simulate the performance of a cluster of size $n$ with a batch size of 32 per device, we take the minimum over $n$ samples from this distribution. Repeating this process to generate many samples allows us to estimate the expected speed-up for any given cluster size. Unfortunately, we cannot make any statistical inference about the processing times of individual sequences since the data is only provided at the granularity of 32 sequences per batch, and it is not clear how much of the computation is done in parallel and how much in serial.

## C  ABLATION STUDY

So far, we have shown that with the introduced adjustments, we can match the accuracy of unpacked BERT. In the following, we wanted to analyze in how far the masking adjustment is required. In Figure 7, we can see that without our adjustments, training loss and accuracy worsen drastically and a longer training time does not lead to a recovery. When not adjusting the positional embedding, the loss and accuracy almost match. However, the accuracy stalls at $71.8\%$ and does not reach the target accuracy of $72.1\%$. So overall, both adjustments are crucial.

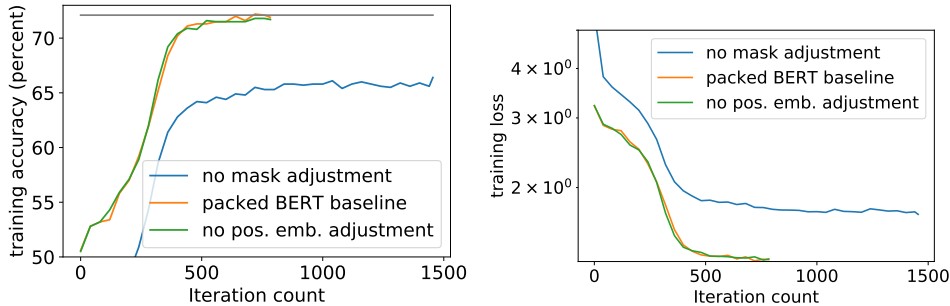

Figure 7: Comparison of learning curves with and without mask or positional embedding adjustment in our packed BERT approach. The grey accuracy baseline to reach is $72.1\%$.

## D  SQUAD 1.1

Packing slightly violates the i.i.d. assumption of data. Thus, it is of interest, if the algorithm matches downstream performance. This is especially relevant with a full training setup without a starting checkpoint. We trained Phase 1&2 of BERT base with and without packing. To avoid giving an advantage to packing by further hyperparameter tuning, we instead reduced the gradient accumulation count for the packed BERT training for Phase 1 and Phase 2 to match the total number of sequences that get processed. With this approach, we could use the same hyperparameters and number of training steps. This gives a slight disadvantage to the packed run. For Phase 2, we used sequence length 348 since longer range attention is not relevant for SQuAD 1.1. For the fine-tuning training on SQuAD 1.1, we did not use packing. After 10 repetitions, the results showed that on average, the F1 score is reduced by $0.003\%$ whereas the EM score is improving by $0.049\%$.

# E  TECHNICAL BACKGROUND ON PACKING

## E.1  CANONICAL PACKING PROBLEM

The bin-packing problem deals with the assignment of items into bins of a fixed capacity such that the number of utilized bins is minimized. In the canonical formulation of the packing problem a vector $s(i)$ of length $n$ is used to represent the items being packed, where $s(i)$ denotes the length of the i-th sequence/item. The allocation of items into bins is tracked through the assignment matrix $B$, where $B_{ij} \in \{0, 1\}$ states whether the i-th sequence should be placed into the j-th bin. In the worst case scenario, every item is assigned to its own bin, thus $B \in \mathbb{R}^{n \times n}$. Notably, $s$ grows linearly in the number of sequences/items being packed and $B$ grows with the square. To mask out unused bins $y_j \in \{0, 1\}$, denotes whether the j-th bin is being used. The optimization objective is to minimize the sum of $y_j$ while making sure to assign each $s_i$ to exactly one bin and not exceeding the maximum bin capacity $s_m$ for each bin. This problem formulation is well known as bin-packing (Korte & Vygen, 2012).

$$
\begin{aligned}
\min_{y \in \{0,1\}^n, B \in \{0,1\}^{n \times n}} \quad & \sum_{j=1}^{n} y_j && \text{Minimize the number of bins.} \\
\text{s.t.} \quad & \sum_{j=1}^{n} b_{ij} = 1 \quad \forall i && \text{Assign each length/sequence to only one bin.} \\
& \sum_{i=1}^{n} s(i) b_{ij} \leq s_m y_j \quad \forall j && \text{Cumulative length cannot exceed capacity.}
\end{aligned}
$$

$$(15)$$

Bin-packing is a strongly NP-complete (Korte & Vygen, 2012) problem. Producing an exact and optimal solution is possible with a variety of existing algorithms, for example with the branch-and-cut-and-price algorithm (Belov & Scheithauer, 2006). However, given that we want to apply it for very large $n$ (16M for the Wikipedia dataset) an approximate approach is required.

## E.2  APPROXIMATE BIN-PACKING PROBLEM

Approximate packing approaches are divided into online and offline algorithms (Johnson, 1973). Online algorithms process incoming sequences one-by-one in a streaming fashion, whereas offline algorithms have a holistic view of all samples to be packed but typically still operate on a per sample basis. This results in best case time and memory complexities of at least $O(n \log(n))$ and solutions that can sometimes be far from optimal, especially for the online algorithms which do not have access to a holistic view of the datasets. The simplest online approach (next-fit) would be to keep a single open bin at any given time. An incoming sequence is added to this open bin if it fits, otherwise the bin is closed (can never be appended to again) and a new one is opened to accommodate the new sequence (Johnson, 1973). In the case of the Wikipedia pre-training dataset almost $25\%$ of the sequences are of length $512$, which makes this approach very inefficient since bins would frequently be closed because the incoming sequence did not fit. More specifically, this approach is not able to efficiently combine one long sequence with one shorter sequence, when the number of long sequences is large. The algorithms that come closest to the approaches proposed in this paper are the online harmonic-k algorithm (Lee & Lee, 1985), which creates harmonic sized bins for the assignment decision, and the offline Modified First Fit Decreasing method (Johnson & Garey, 1985; Yue & Zhang, 1995), which sorts the data, groups it into $4$ size categories and defines a strategy adjusted to these sizes.

In our approaches, we make three major simplifications. We make the problem of bin packing less dependent on $n$ by operating on the histogram of sequence lengths with bin size 1. Hence, we replace $s(i)$ by its histogram $b$ and the bin assignment $y, B$ by a mixture of strategies $x$, where the set of all available packing strategies is modeled as the matrix $A$ (see also Section E.4.2).

Then, we do not solve the full packing problem but focus on a fixed packing depth (in other words the well known 3-partition problem). Last but not least, we solve the limited depth packing problem only approximately either with a non-negativity-constrained linear least squares (Bro & De Jong,

1997) (NNLS) followed by rounding to nearest integer solution or by applying Worst-Fit (Johnson & Garey, 1985; Yue & Zhang, 1995) to the histogram, sorted from largest to smallest (in contrast to using an unsorted dataset). An exact solution would not be appropriate, since the 3-partition problem is strongly NP-complete (Garey & Johnson, 1990) as well.

### E.3 DEFINITIONS

In this section, we standardize the terms used throughout our methods. Firstly, the terms *pack* and *bin* may be used interchangeably. Secondly, the presented packing schemes impose a limit on how many sequences can be packed into any given bin. This limit is referred to as the maximum *packing depth*. For simplicity, we require the different sequence lengths in a pack to always add up exactly to the bin capacity $s_m$ (we can always generate a padding sequence of just the right length to fill-up the bin). A *packing strategy* is a sorted list of sequence lengths, for example $[5, 7, 500]$, such that the total sequence length is no more than $s_m$ and the number of sequences in the pack does not exceed the maximum *packing depth*. The output of a packing scheme is typically as set of *packing strategies* and the corresponding *repeat count* for each strategy stating how many times each strategy should be repeated in order to cover the entire dataset. The strategy *repeat count* is also referred to as the *mixture* of strategies. The objective of the packing algorithm is to jointly design a set of packing strategies and their repeat counts, such that the amount of *padding* is (approximately) minimized. The presence of *padding* in the packs can either be implicit or explicit. For instance for $s_m = 512$ the strategy $[2, 508]$ has an implicit padding of 2 (needed to fill the pack up to the $s_m$). Alternatively, the strategy repeat count may over-subscribe a particular sequence length leading to explicit packing. For instance constructing a pack of $[4, 508]$ may require a new *padding* sequence of length 4 be constructed, if there are not enough sequences of that length in the dataset. The packing algorithms, we present, use both representations.

### E.4 NON-NEGATIVE LEAST SQUARES HISTOGRAM-PACKING

The first algorithm proposed in this paper is suitable for settings where it is desirable to achieve a high packing efficiency with a limited packing depth. The algorithm is deterministic and has three major components described in Sections E.4.1, E.4.2 and E.4.3.

#### E.4.1 ENUMERATING PACKING STRATEGIES OF FIXED PACKING DEPTH

Listing all unique ways of packing up to a maximum *packing depth* can be achieved through dynamic programming. We only consider packing at most up to 3 sequences per pack. This is the smallest packing depth that can eliminate the need for most padding on the Wikipedia dataset. Increasing the depth to 4, increases the size of the packing problem drastically and yields no throughput benefit [2]. With only two sequences, packing would be not as efficient since the distribution on sequence length is not symmetric. We use dynamic programming to enumerate all feasible ways/strategies that up to $M$ sequences of length $1 - 512$ can be packed into a bin of length $512$. For example, a packing strategy may be $[512]$ or $[6, 506]$ or $[95, 184, 233]$. To avoid listing the same strategy multiple times, we enforce the sequence lengths within a pack to occur in sorted order, for example, $[95, 184, 233]$ is equivalent to $[184, 95, 233]$ and should only be listed once. This reduces the search space as well as the space of potential solutions by a factor of 6 approximately and thus significantly accelerates the optimization process. If you had the same strategy repeated 6 times instead of having just one instance of that strategy with weight $X$, you will have six instances with weight $x/6$ (for example, or any other distribution). This would conflict with integer rounding of the solutions and with convergence of optimization algorithms.

#### E.4.2 CONSTRUCTING THE PACKING MATRIX

The number of rows in the packing matrix is equal to the number of different sequence length categories. For instance, if we are using a granularity of 1 token to distinguish between different sequence lengths, then there are "maximum sequence length" rows. Each column of the matrix corresponds to a valid packing strategy (given the depth of packing). An example packing matrix for fitting up to 3 sequences into sequence length 8 is given in Table 2. Each column of the matrix

---

[2]For data distributions that are more skewed than Wikipedia this might look different.

represents a packing strategy. For instance, the first column represents the strategy [1, 1, 6] of packing two length-1 sequences and one length-6 sequence together to form a pack of length 8. The number of strategies (and columns in the matrix) is discussed in Section F. For a packing depth of 3 and maximum sequence length, we obtain around $\frac{s_m^2 + 6s_m + 12}{12}$ strategies. For depth 4, around $\frac{s_m(s_m+4)(2s_m+1)}{288}$ more get added.

Table 2: Example packing matrix for sequence length $8$. Columns represent different kinds of packs. Rows represent the number of sequences in these packs with a certain length. The last column represents a pack with only a single sequence of length six.

| 2 | 1 | 1 | 1 | 0 | 0 | 0 | 0 | 0 | 0 |
|---|---|---|---|---|---|---|---|---|---|
| 0 | 1 | 0 | 0 | 2 | 1 | 1 | 0 | 0 | 0 |
| 0 | 0 | 1 | 0 | 0 | 2 | 0 | 1 | 0 | 0 |
| 0 | 0 | 1 | 0 | 1 | 0 | 0 | 0 | 2 | 0 |
| 0 | 1 | 0 | 0 | 0 | 0 | 0 | 1 | 0 | 0 |
| 1 | 0 | 0 | 0 | 0 | 0 | 1 | 0 | 0 | 0 |
| 0 | 0 | 0 | 1 | 0 | 0 | 0 | 0 | 0 | 0 |
| 0 | 0 | 0 | 0 | 0 | 0 | 0 | 0 | 0 | 1 |

### E.4.3 SOLUTION OF THE NNLS APPROXIMATE PACKING PROBLEM

A solution of the packing problem is the mixture of packing strategies $x$ that minimizes the amount of padding in the packed dataset. We solve directly for the mixture (positive real numbers) and recover the padding as the negative portion of the residual (see Section E.4.4).

$$\min_{x \in \mathbb{R}^m} \quad \|A \cdot x - b\|^2$$
$$\text{s.t.} \ \ x \geq 0 \tag{16}$$

The solution vector $x$ will represent the mixture of the columns of $A$, in other words the mixture of valid packing strategies such that $A \cdot x$ is as close as possible (in the least squares sense) to the histogram of sequence lengths $b$. We obtain a solution with a non-negative least squares implementation (Lawson & Hanson, 1995; Virtanen et al., 2020) Interestingly in the case of sequence length $512$ only 634 out of the 22102 available packing strategies of depth up to 3 are used ($3\%$).

### E.4.4 PADDING AS THE RESIDUALS OF THE PACKING PROBLEM

We compute the residuals of the least squares solution (after rounding the mixture to integer) as:
$$r = b - A \cdot round(x) \tag{17}$$
The negative portion of the residuals represents sequences that we are "short". That is, there is a deficit of those sequences and we are over-subscribing to them. The positive portion of the residuals represents sequences which have failed to be packed. Typically, there is a deficit of short sequences and a surplus of long sequences as demonstrated by the following plot.

In total, there are $n = 16'279'552$ sequences in the Wikipedia pre-training dataset. After the non-negative least squares packing (and rounding to integer solution) there are $56'799$ un-packed sequences left un-packed (about $0.352\%$). The residual on sequence lengths 1 to 8 are $[-4620, -4553, -4612, -4614, -3723, -3936, -3628, -3970]$. These negative residuals imply that we need to add this many sequences of their corresponding sequence length to realize the mixture of packing strategies. In total the first iteration introduces $7.94 10^6$ tokens of padding. In contrast large sequence lengths have a positive residual (a surplus of unused sequences). For sequence lengths 504 to 512 the values are $[3628, 3936, 3724, 4613, 4612, 4553, 4619, 0]$. Note that sequence length 512 has a residual of 0 since they do not need packing. Intermediate sequence lengths typically have non-zero (but much smaller) residuals.

The detailed code for the algorithm is provided in Listing 2.

### E.4.5 RESIDUAL WEIGHTING

A natural extension of the non-negative least squares problem introduced in Section E.4.3 is to weight the residuals on different sequence length differently.

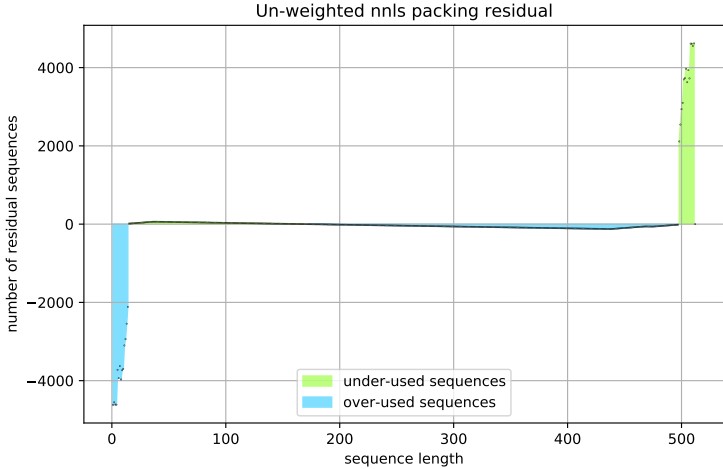

Figure 8: Visualization of the residual of the NNLS packing problem

$$\min_{x \in \mathbb{R}^m} \quad \|(wA) \cdot x - (wb)\|^2$$
$$\text{s.t.} \quad x \geq 0 \tag{18}$$

We should not significantly penalize a deficit in short sequence lengths (smaller than 8 tokens) as adding up to 8 tokens of padding is not much overhead. Similarly, a surplus in long sequences is not worrisome because the amount of padding needed to achieve a sequence length of 512 is small. Reducing the weight of the residual on the first 8 tokens to 0.09 leads to the following residual plot shown on the right in Figure 9. In this case the residual is almost entirely shifted to the shorter sequences and the positive residual on the longer sequences has virtual disappeared.

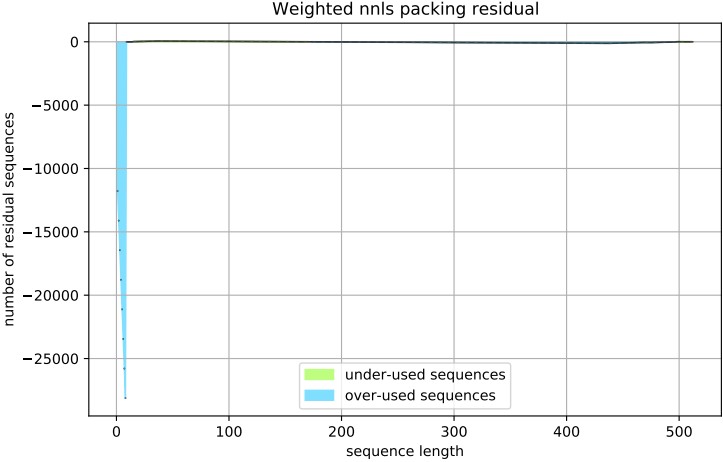

Figure 9: Visualization of the weighted residual of the NNLS packing problem

### E.5 DISCUSSION OF RESIDUAL WEIGHT CHOICE

This section discusses the choice and effect of the weighting parameters in the NNLSP packing algorithm. To simplify the problem of selecting reasonable defaults for the residual weights, we use just two parameters to completely describe the weights: an "offset" parameter and a "weight" parameter. Originally, all sequence length residuals are given the same weight of 1. This results in a packing with leftover long sequences, because there are not enough short sequences to pack them with. To reduce the residual on long sequences, we could either increase the residual weight on long sequences or reduce the weight on short sequences. We chose to reduce the weight on short sequences. Specifically, sequence lengths up to the "offset" length have a reduced "weight". The other residual weights stay at 1.

To start, we chose an offset of 8 tokens, which is the smallest power of 2 for which there are examples in the Wikipedia dataset. We decrease the weight on sequences shorter than the "offset" from 1 to 0.9 to 0.09 to see which order of magnitude is the most appropriate. On visual inspection (looking at the residual plots as in Figure 9), we found that 0.9 still left too many long sequences unpacked. So, we reduced the weight a further order of magnitude to 0.09. This seemed sufficient to encourage nearly all long sequences to pack. While visual inspection helps in understanding how many long/short sequences are leftover, we are also interested in the impact the weights have on the overall efficiency of the packing.

Without any weighting, we get $99.746359\%$ efficiency, whereas the weighted approach results in $99.746274\%$ efficiency. Hence, we conclude that the impact of the weights on the packing efficiency is very limited. Additionally, using an "offset" length of 4, resulted in similar numbers, for the full range of weights from 0 to 1. Using a weight of 0 for an "offset" length of 8 resulted in insignificantly higher efficiency of $99.7519\%$, whereas using an "offset" length of 16 reduces performance to $99.38964\%$. A weight of 0 implies that the residual on those lengths can be safely ignored, i.e., the packing algorithm can thus add as many short sequences as it chooses without any penalty. It is very interesting that this does not significantly impact the packing efficiency, and can even have a slightly positive impact. However, increasing the "offset" length further significantly decreases the performance with weight 0. Keeping the weight at 0.09 and increasing the length reduces performance slightly, for example with $99.53\%$ at length 256 and $99.728\%$ at length 16.

For our Squad analysis, weighting improved the efficiency slightly from $96.94\%$ to $97.38\%$. Fine tuning further with direction grid search, delivered a local optimum of $98.767\%$ efficiency with length 64 and weight 0.002.

Overall the influence of different residual weights on the packing efficiency (and the acceleration factor) is less than $1\%$. This might differ from application to application, but it shows that we are able to use the residual weights to achieve secondary targets (like not having leftover long sequences) without significantly compromising the packing efficiency.

# F COMPLEXITY ANALYSIS OF THE PROPOSED PACKING APPROACHES

Since approximate packing algorithms have a complexity of at least $O(n \log(n))$ and we would like to be able to tackle datasets with 2K million samples, we will discuss the complexity of our packing algorithms in this section. The complexity depends on the maximum sequence length $s_m$, the number of samples $n$, and the packing depth $d$.

To create the histogram, we have to iterate over the data once ($O(n)$). Our histograms will be binned by size 1, meaning one bin for each sequence length. Hence, a dictionary can be generated ($O(s_m)$) and used for the sorting ($O(1)$). The respective histogram vector has dimension $s_m$.

## F.1 COMPLEXITY ANALYSIS OF NON-NEGATIVE LEAST-SQUARES HISTOGRAM-PACKING

For a packing depth of one, there is only the strategy $[s_m]$. For a packing depth of two, we add the strategies $[1, s_m - 1], ..., [s_m - \lfloor \frac{s_m}{2} \rfloor]]$ which results in an additional $\lfloor \frac{s_m}{2} \rfloor$ potential strategies. Following the dynamic programming approach, the number of possible additional strategies of depth three can be calculated with

$$
\begin{aligned}
\text{\# potential strategies} &= \sum_{j=1}^{\lfloor \frac{s_m}{3} \rfloor} \sum_{i=j}^{\lfloor \frac{s_m-j}{2} \rfloor} 1 = \sum_{j=1}^{\lfloor \frac{s_m}{3} \rfloor} \left\lfloor \frac{s_m - j}{2} \right\rfloor - (j - 1) \\
&\approx \sum_{j=1}^{\lfloor \frac{s_m}{3} \rfloor} \frac{s_m}{2} - \frac{3}{2}j \approx \frac{s_m}{2}\frac{s_m}{3} - \frac{3}{2}\frac{s_m/3(s_m/3 + 1)}{2} \\
&\approx \left\lceil \frac{s_m^2}{12} \right\rceil
\end{aligned}
\tag{19}
$$

Note that for $s_m = 512$ the approximation is exact. This means that our strategy matrix $A$ has the dimensions $s_m \times \left( \left\lceil \frac{s_m^2}{12} \right\rceil + \lfloor \frac{s_m}{2} \rfloor + 1 \right)$. Overall, this leaves us with a space complexity of $s_m^3$ since $A$ is larger than $w$, $x$, and $b$. So it contains 11'316'224 numbers which is still much smaller than $n$. Note that the original data matrix $B$ had $n^2$ entries, which all needed to be optimized together with the $n$ bin assignments $y$. We now have only $\left\lceil \frac{s_m^2}{12} \right\rceil + \lfloor \frac{s_m}{2} \rfloor$ free variables in the strategy vector $x$. Also note that $A$ can be precomputed when $s_m$ is known and is independent of the number of samples. Given a problem matrix with dimension $i \times j$, Luo et al. (Luo & Duraiswami, 2011) indicate that the asymptotic complexity of most solution approaches is $O(ij^2)$, whereas they propose an $O(ij)$ solution. Since we use the standard SciPy implementation (Lawson & Hanson, 1995), our estimated total time complexity for NNLSHP is $O(n + s_m^5)$.

For $s_m = 2048$, the estimate would be $350'540$ potential strategies which is still far less than the number of samples. For packing depth 4, we calculate (Wolfram Research Inc.):

$$
\begin{aligned}
&\sum_{k=1}^{\lfloor \frac{s_m}{4} \rfloor} \sum_{j=k}^{\lfloor \frac{s_m-k}{3} \rfloor} \sum_{i=j}^{\lfloor \frac{s_m-j-k}{2} \rfloor} 1 \\
&\approx \sum_{k=1}^{\lfloor \frac{s_m}{4} \rfloor} \sum_{j=k}^{\lfloor \frac{s_m-k}{3} \rfloor} \frac{s_m - k + 2 - 3j}{2} \\
&\approx \sum_{k=1}^{\lfloor \frac{s_m}{4} \rfloor} \frac{1}{12}(s + 4 - 4k)(s + 3 - 4k) \\
&\approx \frac{1}{288}s(2s^2 + 9s + 4) \\
&= \frac{1}{288}s(s + 4)(2s + 1)
\end{aligned}
\tag{20}
$$

So with $s_m = 512$, there would be around $940K$ strategies. In our implementation, this number of strategies would be too high to create the problem matrix. One alternatives to simplify would be to not use the exact length of sequences but to only consider even numbers for the sequence length and round up. That way arbitrary sequence length could also be handled and the limiting factor would be the complexity of the attention layer in BERT which does not scale well with the sequence length.

### F.2 Complexity Analysis of shortest-pack-first histogram-packing

The complexity calculation of SPFHP is straightforward. We go over the whole data once for the histogram sorting. Next, we iterate over each of the $s_m$ bins in the histogram. Lastly, we iterate over all strategies that were encountered so far. It can be proven that, at each iteration, the number of strategies can be maximally increased by one. In each step, we potentially add a sequence to existing strategies but a new strategy is opened up only in the final step, when we either create a new strategy or we split one of the existing strategies into two. Hence, the number of strategies is bounded by $s_m$ and the overall time complexity is bounded by $O(n + s_m^2)$. The space complexity is $O(s_m^2)$ since we need to store up to $s_m$ strategies with maximum $s_m$ counts for different sequence length.

## G Performance Comparison to GREEDY Packing in T5

T5 (Raffel et al., 2019) is normally trained on the C4 dataset. However, to give an idea of the difference in packing efficiency and acceleration compared to our newly introduced algorithm, we can analyse the performance of greedy aggregation of samples on our given Wikipedia dataset.

We take the histogram and cast it back to a list of different sequence lengths since this is all that matters for analysing packing behaviour. Next, we randomly shuffle the dataset and iterate with the greedy aggregation algorithm multiple times to account for randomness. We iterate sequence by sequence and combine them provided the maximum sequence length of $512$ is not yet reached. If it is exceeded, the packed sequence is considered finished and a new sequence is started.

The greedy packing algorithm itself takes a bit more than $10$ seconds, since we are operating on single sequences and not histogram counts. The efficiency of this approach is $78.24\%$ (standard deviation of $0.005$) compared to our $99.75\%$ for NNLSHP. The respective acceleration would be around $1.566x$ compared to our $2x$. With respective separator tokens, the performance decreases around $0.13\%$ for one separator token and $0.27\%$ when two separator tokens are required between two sequences. Following the brief documentation at `https://github.com/tensorflow/tensor2tensor/blob/5623deb79cfcd28f8f8c5463b58b5bd76a81fd0d/tensor2tensor/data_generators/generator_utils.py#L1086`, two separator tokens would be expected in the T5 processing.

In addition to the packing preprocessing, our paper proposes, rather than using separator tokens, to instead modify the masking of the attention matrix during training. The RoBERTa paper shows that avoiding contamination of sequences from different documents can consistently improve downstream F1 scores by $0.35\%$.

# H    PACKING SQUAD 1.1

We tokenized SQuAD (Rajpurkar et al., 2016) for BERT (Devlin et al., 2019a) with maximum sequence length $384$ and visualized the histogram over the sequence length (Figure 10). The distribution looks similar to the Wikipedia dataset but is slightly less skewed. However, the maximum sequence length only had an occurrence of $1.2\%$ compared to $23.5\%$. Hence, the theoretical unpadding speedup is 2.232. In Table 3, we can see that SPFHP does not concatenate more than 3 samples and obtains $97.54\%$ efficiency in contrast to a maximally used depth of 16 with $99.60\%$ efficiency on Wikipedia, because of the less skewed distribution. Note that we have less than $90'000$ samples. Hence, NNLSHP is less efficient because the rounding in the residuals has a much larger impact compared to more than 16 million sequences in the Wikipedia dataset.

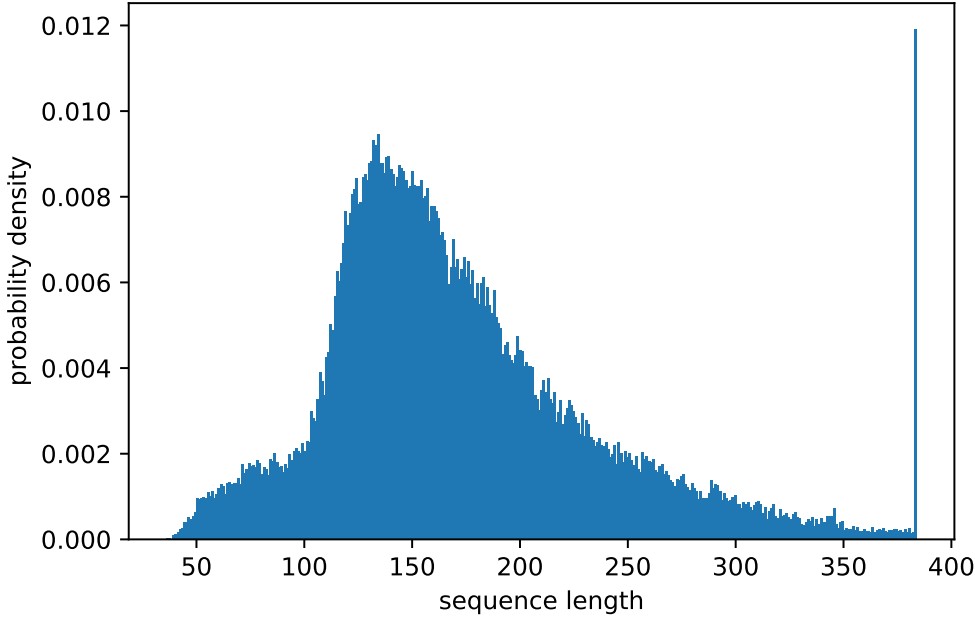

Figure 10:  SQuAD 1.1 BERT pre-training dataset sequence length histogram for maximum sequence length of 384.

Table 3: Performance results of proposed packing algorithms for SQuAD 1.1 BERT pre-training.

| packing depth | packing algorithm | # strategies used | # packs | # tokens | # padding tokens | efficiency (%) | packing factor |
|---|---|---|---|---|---|---|---|
| 1 | none | 348 | 88641 | 34038144 | 18788665 | 44.801 | 1.000 |
| 2 | SPFHP | 348 | 45335 | 17408640 | 2159161 | 87.597 | 1.955 |
| 3 | NNLSHP | 398 | 40808 | 15670272 | 420793 | 97.310 | 2.172 |
| 3/max | SPFHP | 344 | 40711 | 15633024 | 383545 | 97.547 | 2.177 |

# I   PACKING GLUE

To explore a variety of datasets and emphasize that skewed distributions are common, we explored all datasets in the GLUE benchmark (Warstadt et al., 2018; Wang et al., 2018) that came with training data. We loaded the datasets using the HuggingFace dataset loading API (Wolf et al., 2020). For preprocessing, we followed the implementation in the HuggingFace transformers repository (Wolf et al., 2020) [3] and extracted the respective data processing snippets to obtain tokenized data with a maximum sequence length of 128. The histogram of the sequence length for each of the included datasets is displayed in Figure 11 and the packing results are given in Table 4. Each dataset benefits from packing. The lower the mean, the higher the packing factors are that can be reached but with a higher packing depth.

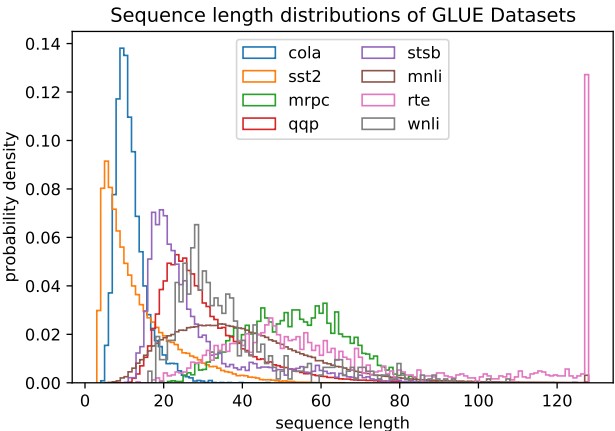

Figure 11: GLUE dataset sequence length histograms for maximum sequence length of 128.

Table 4: Performance results of proposed packing algorithms for the GLUE dataset. Only the baseline and the SPFHP packing results without limiting the packing depth are displayed.

| data name | packing depth | # strategies used | # packs | # tokens | # padding tokens | efficiency (%) | packing factor |
|---|---|---|---|---|---|---|---|
| cola | 1 | 34 | 8551 | 1094528 | 997669 | 8.849 | 1.000 |
| cola | 13/max | 29 | 913 | 116864 | 20005 | 82.882 | 9.366 |
| sst2 | 1 | 64 | 67349 | 8620672 | 7723633 | 10.406 | 1.000 |
| sst2 | 15/max | 64 | 7691 | 984448 | 87409 | 91.121 | 8.757 |
| mrpc | 1 | 77 | 3668 | 469504 | 274214 | 41.595 | 1.000 |
| mrpc | 4/max | 74 | 1606 | 205568 | 10278 | 95.000 | 2.284 |
| qqp | 1 | 123 | 363846 | 46572288 | 35448844 | 23.884 | 1.000 |
| qqp | 5/max | 123 | 97204 | 12442112 | 1318668 | 89.402 | 3.743 |
| stsb | 1 | 85 | 5749 | 735872 | 575993 | 21.726 | 1.000 |
| stsb | 6/max | 83 | 1367 | 174976 | 15097 | 91.372 | 4.206 |
| mnli | 1 | 124 | 392702 | 50265856 | 34636487 | 31.093 | 1.000 |
| mnli | 8/max | 124 | 123980 | 15869440 | 240071 | 98.487 | 3.167 |
| rte | 1 | 112 | 2490 | 318720 | 152980 | 52.002 | 1.000 |
| rte | 4/max | 108 | 1330 | 170240 | 4500 | 97.357 | 1.872 |
| wnli | 1 | 72 | 635 | 81280 | 57741 | 28.960 | 1.000 |
| wnli | 6/max | 63 | 192 | 24576 | 1037 | 95.780 | 3.307 |

---

[3] `https://github.com/huggingface/transformers/blob/master/examples/text-classification/run_glue.py`

## J    PACKING AUDIO DATA (LIBRISPEECH)

In this section, we show that packing can benefit other domains than NLP like ASR. We use the LibiSpeech dataset (Panayotov et al., 2015) and preprocess it as described at a reference implementation [4]. The resulting histograms for the subsampled audio sample lengths and respective text labels are provided in Figure 12.

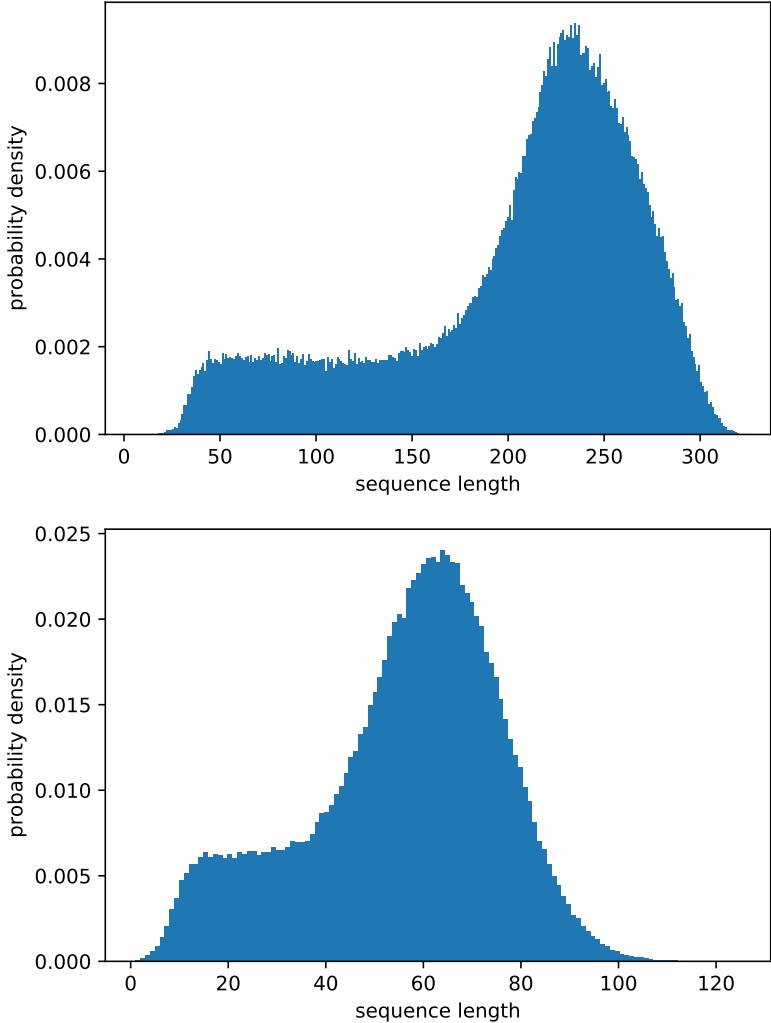

Figure 12: LibriSpeech sequence length histograms of preprocessed audio data [top] as well as target text data [bottom].

It can be seen that the audio sequence length is dominated by long sequences with $38\%$ of required padding to meet the max sequence length of 330. Thus the theoretical optimal speed-up of $1.6x$ cannot be reached. However, $80\%$ efficiency are possible with any of the proposed packing algorithms to achieve $1.3x$ speed-up. This can be already achieved by combining up to 2 sequences. To achieve almost perfect packing efficiency, a sequence length around 457 and concatenating up to 8 sequences is required. Due to the quadratic increased computational load that usually comes with longer sequence length, increasing the sequence length is not practical.

If processing and packing the text data independently of the audio, $99.99\%$ efficiency can be achieved with a speed-up of $2.24x$.

---

[4] https://github.com/mlcommons/training/tree/master/rnn_speech_recognition/pytorch

# K PACKING PAPER ABSTRACTS (PUBMED)

This section analyses the length of abstracts to give an intuition about how different documents can be in length. Figure 13 depicts the length of abstracts in characters extracted from PubMed [5]. If these abstracts were directly used as sequences, a character length of 1000 could result in $1.9x$ speed-up from packing. The potential speed-ups for length 2000, 3000, 4000 would be $2x$, $3x$, and $4x$, respectively. Note that, document clean-up procedures would usually eliminate documents that are too short or too long for data sanitizing purposes.

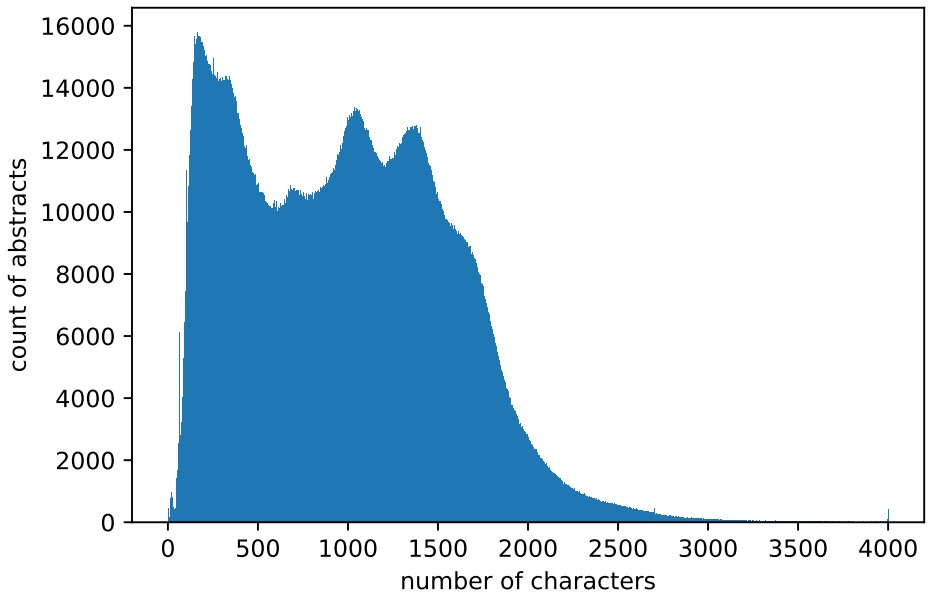

Figure 13: Abstract length distribution in PubMed.

Note that for the processing in BlueBERT (Peng et al., 2019), paper titles and abstracts get separated into sequences, tokenized, and then combined with the BERT sequence combination approach for a maximum sequence length of 128 tokens. Thus, it results in a different distribution.

---

[5]https://huggingface.co/datasets/pubmed

## L FURTHER LEARNING CURVES

This section provides further learning curves related to Section 4.2.

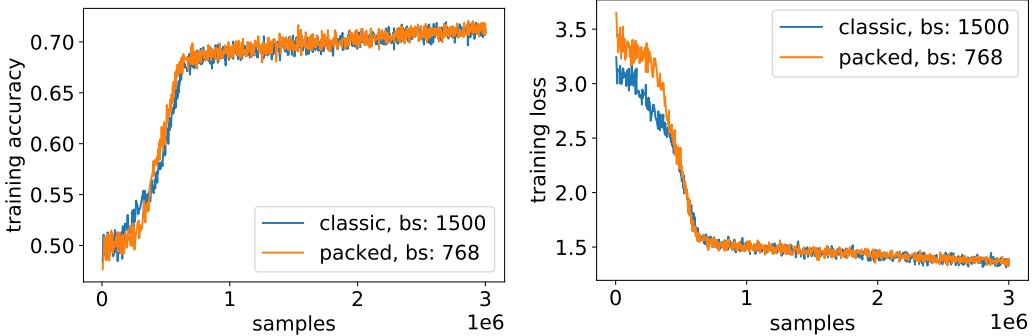

Figure 14: Comparison of learning curves for packed and unpacked processing with **reduced batch size** for the packed approach.

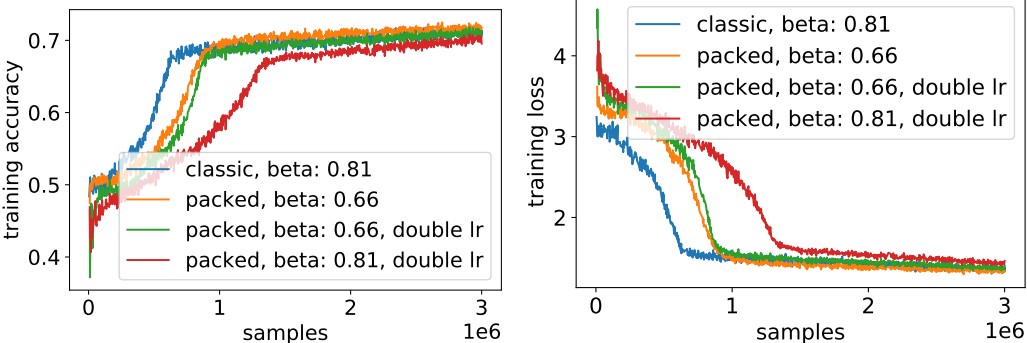

Figure 15: Comparison of learning curves for packed and unpacked processing with **heuristics** applied.

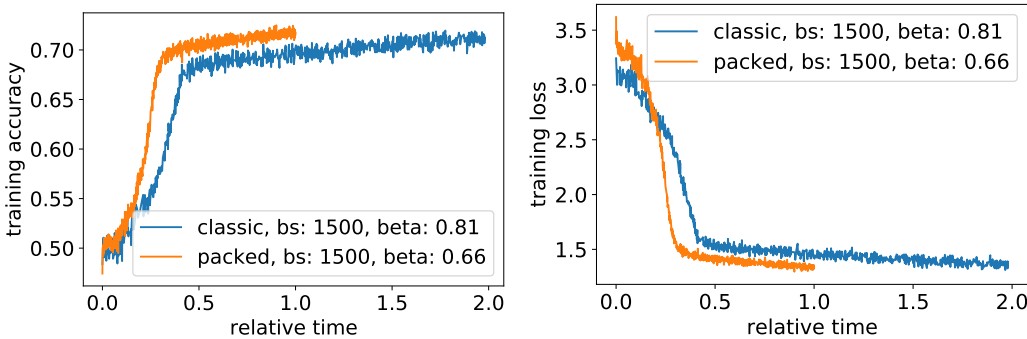

Figure 16: Comparison of learning curves for packed and unpacked processing in the **optimized setup**.

## M   FINE-TUNED LONGEST-PACK-FIRST HISTOGRAM-PACKING

In the main paper, we focused on SPFHP due its simplicity. In this section, we analyse the effect of applying the "Best-Fit" algorithm (Johnson, 1973). Here, the longest pack that still fits the sequence is chosen instead of the shortest one. In contrast to SPFHP, we additionally consider splitting the histogram count, if it can fit multiple times. A simple example is sequence length 256, where we divide the respective histogram count by 2 to create the optimal pack with strategy $[256, 256]$ instead of the strategy $[256]$. This latter strategy would be complemented by other sequences but would probably not result in an optimal packing. The implementation of this approach is much more complex than the SPFHP implementation. The code is provided in Listing 8 and the results in Table 5.

| pack. depth | # strat. used | # packs | # tokens | # padding tokens | efficiency (%) | pack. factor |
|---|---|---|---|---|---|---|
| 1 | 508 | 16279552 | 8335130624 | 4170334451 | 49.967 | 1.000 |
| 2 | 634 | 10099081 | 5170729472 | 1005933299 | 80.546 | 1.612 |
| 3 | 648 | 9090154 | 4654158848 | 489362675 | 89.485 | 1.791 |
| 4 | 671 | 8657119 | 4432444928 | 267648755 | 93.962 | 1.880 |
| 8 | 670 | 8207569 | 4202275328 | 37479155 | 99.108 | 1.983 |
| 16 | 670 | 8140006 | 4167683072 | 2886899 | 99.931 | 2.000 |
| 29/max | 670 | 8138483 | 4166903296 | 2107123 | 99.949 | 2.000 |

Table 5: Performance results of longest-pack-first histogram-packing for Wikipedia BERT pre-training with maximum sequence length 512.

We can see that longest-pack-first histogram-packing (LPFHP) uses a much higher packing depth when no limit is set (29 instead of 16). Splitting the histogram counts results in slightly higher numbers of used strategies compared to SPFHP where the number of used strategies is limited by the maximum sequence length. The best efficiency of LPFHP is $99.949\%$ with packing factor of 2 which is slightly higher than the $99.75\%$ (1.996 packing factor) for NNLSHP and $99.6\%$ for SPFHP (1.993 packing factor). All algorithms are very close to the upper limit.

Note that for NNLSHP, we only fill up the unpacked samples with padding. Applying best-fit on the remains, similar results can be expected. Although the benefits of the improved algorithm are negligible, we share the concept and code below in case packing is applied to other data with a different distribution that would benefit more from it, or for applications where only perfectly packed sequences without padding are of interest.

## N    EXTENDED NNLS WITH PADDING TOKEN WEIGHTING

In Section E.4.4, we defined the residual as

$$r = b - A \cdot round(x) \tag{21}$$

and discovered that a positive residual corresponds to sequences that we did not pack at all and should be avoided. Negative residuals correspond to padding and should be minimized. Due to this discrepancy, we decided to set small weights for very short sequences (that don't occur in the data). However, it was not possible to directly optimize the amount of padding. A negative residual component for length $i$, $r_i$, results in $|r_i| \cdot i$ padding tokens, however a positive residual actually results into $(512 - r_i) \cdot i$ padding tokens. This cannot be addressed by our weighting approach in

$$\min_{x \in \mathbb{R}^m} \quad \|(wA) \cdot x - (wb)\|^2$$
$$\text{s.t.} \quad x \geq 0 \tag{22}$$

Working within the NNLS approach, we can strictly enforce a non-positive residual $r$ (before rounding to integer). To that end, we define a new auxiliary variable $\overline{r} \approx -(b - Ax)$ which is the negative of the residual, $r$. This will allow us to reformulate the objective $r \leq 0$ to the non-negative constraint: $\overline{r} \geq 0$.

$$\min_{x \in \mathbb{R}^m} \quad \|(wA) \cdot x - (wb)\|^2 + \|\overline{w} \cdot A \cdot x - \overline{w} \cdot b - \overline{w} \cdot \overline{r}\|^2$$
$$\text{s.t.} \quad x \geq 0$$
$$\overline{r} \geq 0 \tag{23}$$

This will enforce $\overline{r} = Ax - b \geq 0$ due to the large weight, $\overline{w} := 10^6$, and no upper limits on $\overline{r}$. Now, we can set $w_i := i$ to optimize for the padding tokens. Due to the use of the squared error, we would however optimize the squared sum of padding tokens instead of the preferred sum of padding tokens. To accomplish the latter, we would have to replace the L2-norm problem by an L1-norm problem which would be too complex to solve. Note that due to rounding, the unwanted positive residuals $r$ ($\overline{r} < 0$) might still occur. This could be avoided by rounding up $x$ instead of normal rounding of $x$. To put the new formulation into a solver, we replace

$$b \text{ by } \begin{pmatrix} b \\ b \end{pmatrix}, \ x \text{ by } \begin{pmatrix} x \\ r \end{pmatrix}, \ w \text{ by } \begin{pmatrix} w \\ \overline{w} \end{pmatrix}, \text{ and } A \text{ by } \begin{pmatrix} A & 0_m \\ A & -D_m \end{pmatrix}, \tag{24}$$

where $0_m$ is an $m \times m$ matrix with $m$ being the maximum sequence length, $512$, and $D_m$ is a unit matrix of the same dimensions as $0_m$. Since, we are already close to optimum especially on the Wikipedia dataset, the results are only a little bit better. The processing time however increases from $30$ to $415$ seconds without considering the increased time for constructing the processing matrix. Since the slightly improved algorithm might be nevertheless relevant for other applications, we share it in Listing 9.

## O  PACKING SOURCE CODE

Listing 2: Non-negative least squares histogram-packing

```python
import time
import numpy as np
from scipy import optimize, stats
from functools import lru_cache

def get_packing_matrix(strategy_set, max_sequence_length):
    num_strategies = len(strategy_set)
    A = np.zeros((max_sequence_length, num_strategies), dtype=np.int32)
    for i, strategy in enumerate(strategy_set):
        for seq_len in strategy:
            A[seq_len - 1, i] += 1
    return A

@lru_cache(maxsize=None)
def get_packing_strategies(start_length, minimum_increment, target_length, depth):
    gap = target_length - start_length
    strategies = []
    # Complete the packing with exactly 1 number
    if depth == 1:
        if gap >= minimum_increment:
            strategies.append([gap])
    # Complete the sample in "depth" steps, recursively
    else:
        for new in range(minimum_increment, gap + 1):
            new_gap = target_length - start_length - new
            if new_gap == 0:
                strategies.append([new])
            else:
                options = get_packing_strategies(start_length + new, new, target_length, depth - 1)
                for option in options:
                    if len(option) > 0:
                        strategies.append([new] + option)
    return strategies

def pack_using_nnlshp(histogram, max_sequence_length, max_sequences_per_pack):
    # List all unique ways of packing to the desired maximum sequence length
    strategy_set = get_packing_strategies(0, 1, max_sequence_length, max_sequences_per_pack)
    print(f"Packing will involve {len(strategy_set)} unique packing strategies.")
    # Get the packing matrix corresponding to this list of packing strategies
    A = get_packing_matrix(strategy_set, max_sequence_length)
    # Weights that penalize the residual on short sequences less.
    penalization_cutoff = 8
    w0 = np.ones([max_sequence_length])
    w0[:penalization_cutoff] = 0.09
    # Solve the packing problem
    print(f"Sequences to pack: ", histogram.sum())
    start = time.time()
    strategy_repeat_count, rnorm = optimize.nnls(np.expand_dims(w0, -1) * A, w0 * histogram)
    print(f"Solving non-negative least squares took {time.time() - start:3.2f} seconds.")
    # Round the floating point solution to nearest integer
    strategy_repeat_count = np.rint(strategy_repeat_count).astype(np.int64)
    # Compute the residuals, shape: [max_sequence_length]
    residual = histogram - A @ strategy_repeat_count
    # Handle the left-over sequences i.e. positive part of residual
    unpacked_seqlen = np.arange(1, max_sequence_length + 1)[residual > 0]
    for l in unpacked_seqlen:
        strategy = sorted([l, max_sequence_length - l])  # the depth 1 strategy
        strategy_index = strategy_set.index(strategy)
        strategy_repeat_count[strategy_index] += residual[l-1]
    # Re-compute the residual with the updated strategy_repeat_count
    # This should now be strictly < 0
    residual = histogram - A @ strategy_repeat_count
    # Add padding based on deficit (negative residual portion of residual)
    padding = np.where(residual < 0, -residual, 0)
    # Calculate some basic statistics
    sequence_lengths = np.arange(1, max_sequence_length + 1)
    old_number_of_samples = histogram.sum()
    new_number_of_samples = int(strategy_repeat_count.sum())
    speedup_upper_bound = 1.0/(1 - (histogram*(1 - sequence_lengths / max_sequence_length)).sum()/
      old_number_of_samples)
    num_padding_tokens_packed = (sequence_lengths * padding).sum()
    efficiency = 1 - num_padding_tokens_packed/(new_number_of_samples*max_sequence_length)
    print(f"Packing efficiency (fraction of real tokens): {efficiency:3.4f}\n",
          f"Speed-up theoretical limit: {speedup_upper_bound:3.4f}\n",
          f"Achieved speed-up over un-packed dataset: {old_number_of_samples/new_number_of_samples:3.5f}")
    return strategy_set, strategy_repeat_count
```

Listing 3: Shortest-pack-first histogram-packing

```python
from collections import defaultdict
import numpy as np

def add_pack(pack, count, tmp, final, limit, offset):
    """Filter out packs that reached maximum length or number of sequences."""
    if len(pack) == limit or offset == 0:
        final[offset].append((count, pack))
    else:
        tmp[offset].append((count, pack))

def pack_using_spfhp(histogram, max_sequence_length, max_sequences_per_pack):
    """Shortest-pack-first histogram-packing algorithm."""
    reversed_histogram = np.flip(histogram)
    # Initialize main strategy data dictionary.
    # The key indicates how many tokens are left for full length.
    # The value is a list of tuples, consisting of counts and respective packs.
    # A pack is a (sorted) list of sequence length values that get concatenated.
    tmp_strategies_per_length = defaultdict(list)
    strategies_per_length = defaultdict(list)
    # Index i indicates here, how much space is left, due to reversed histogram
    for i in range(max_sequence_length):
        n_sequences_to_bin = reversed_histogram[i]
        length_to_bin = max_sequence_length - i
        offset = i + 1  # largest possible offset
        while n_sequences_to_bin > 0:
            if (length_to_bin + offset) in tmp_strategies_per_length:
                # extract shortest pack that will get modified
                n_sequences_to_pack, pack = tmp_strategies_per_length[
                    length_to_bin + offset].pop()
                new_pack = pack + [length_to_bin]
                count = min(n_sequences_to_pack, n_sequences_to_bin)
                if n_sequences_to_pack > n_sequences_to_bin:
                    # old pack gets reduced
                    n_sequences_to_pack -= n_sequences_to_bin
                    tmp_strategies_per_length[length_to_bin + offset].append(
                        (n_sequences_to_pack, pack))
                    n_sequences_to_bin = 0
                else:
                    n_sequences_to_bin -= n_sequences_to_pack
                add_pack(new_pack, count,
                         tmp_strategies_per_length, strategies_per_length,
                         max_sequences_per_pack, offset)
                # clean up to speed up main key search
                if not tmp_strategies_per_length[length_to_bin + offset]:
                    tmp_strategies_per_length.pop(length_to_bin + offset)
            else:
                offset -= 1
            # Does not fit anywhere. Create new pack.
            if offset < 0:
                add_pack([length_to_bin], n_sequences_to_bin,
                         tmp_strategies_per_length, strategies_per_length,
                         max_sequences_per_pack, i)
                n_sequences_to_bin = 0
    # merge all strategies
    for key in tmp_strategies_per_length:
        strategies_per_length[key].extend(tmp_strategies_per_length[key])
    # flatten strategies dictionary
    strategy_set = []
    strategy_repeat_count = []
    for key in strategies_per_length:
        for count, pack in strategies_per_length[key]:
            pack.reverse()
            strategy_set.append(pack)
            strategy_repeat_count.append(count)
    return strategy_set, np.array(strategy_repeat_count)
```

Listing 4: Evaluation function of shortest-pack-first histogram-packing

```python
"""Max depth analysis of shortest-pack-first histogram-packing."""
from collections import defaultdict
import tabulate
import time
import numpy as np

def evaluate_spfhp(histogram, max_sequence_length):
    """Evaluate shortest-pack-first histogram-packing algorithm."""
    stats_data = [["pack. depth", "# strat. used", "# packs", "# tokens",
                   "# padding tok.", "efficiency (%)", "pack.factor", "time"]]
    for max_sequences_per_pack in [1, 2, 3, 4, 8, 16, "max"]:
        start = time.time()
        strategy_set, strategy_repeat_count = pack_using_spfhp(
            histogram, max_sequence_length, max_sequences_per_pack)
        duration = time.time() - start

        # Performance Evaluation of packing approach
        n_strategies = int(len(strategy_set))
        packs = int(sum(strategy_repeat_count))
        sequences = sum([count*len(pack) for count, pack in
                         zip(strategy_repeat_count, strategy_set)])
        total_tokens = int(max_sequence_length * packs)
        empty_tokens = int(sum([
            count*(max_sequence_length-sum(pack)) for count, pack in
            zip(strategy_repeat_count, strategy_set)]))
        token_efficiency = 100 - empty_tokens / total_tokens * 100
        if max_sequences_per_pack == "max":
            m_length = max([len(pack) for pack in strategy_set])
            max_sequences_per_pack = "max ({})".format(m_length)
        stats_data.append([
            max_sequences_per_pack, n_strategies, packs, total_tokens,
            empty_tokens, token_efficiency, sequences / packs, duration])
    print(tabulate.tabulate(stats_data, headers="firstrow", floatfmt=".3f"))
```

Listing 5: Loss calculation

```python
# The number of sequences in each batch may vary
sequences_in_batch = tf.reduce_sum(tf.reduce_max(masked_lm_weight, -1))
sequences_in_batch = tf.cast(sequences_in_batch, tf.float32)
# Create the 0/1 mask that will be used to un-packed sequences
masked_lm_weight = tf.reshape(masked_lm_weight, [B, 1, -1])
sequence_selection = tf.reshape(tf.range(1, max_sequences_per_pack + 1), [1, -1, 1])
sequence_selection = tf.cast(masked_lm_weight == sequence_selection, tf.float32)
# Apply the mask to un-pack the loss per sequence
nll_per_token = tf.reshape(nll_per_token, [B, 1, -1])
nll_per_sequence = sequence_selection * nll_per_token
# Normalize the per-sequence loss by the number of mlm-tokens in the sequence (as is standard)
attempted = tf.reduce_sum(sequence_selection, -1, keepdims=True)
attempted = attempted + tf.cast(attempted == 0, tf.float32)  # prevent NaNs when dividing by attempted
nll_per_sequence = nll_per_sequence/attempted
# Average per-batch loss (so contributions from different batches are comparable)
lm_loss = tf.reduce_sum(nll_per_sequence)/sequences_in_batch
```

Listing 6: Wikipedia and SQuAD 1.1 histograms

```python
import numpy as np
wikipedia_histogram = np.array([
    0, 0, 0, 0, 1821, 1226, 1969, 1315, 1794, 1953, 3082, 3446, 4166, 5062,
    9554, 16475, 19173, 17589, 17957, 19060, 21555, 23524, 26954, 30661, 33470, 36614, 40134, 43256,
    46094, 49350, 52153, 55428, 58109, 60624, 63263, 64527, 65421, 66983, 68123, 68830, 70230, 70486,
    72467, 72954, 73955, 74311, 74836, 74489, 74990, 75377, 74954, 75096, 74784, 74698, 74337, 74638,
    74370, 73537, 73597, 73153, 72358, 71580, 71082, 70085, 69733, 69445, 67818, 67177, 66641, 65709,
    64698, 63841, 63218, 62799, 61458, 60848, 60148, 59858, 58809, 58023, 56920, 55999, 55245, 55051,
    53979, 53689, 52819, 52162, 51752, 51172, 50469, 49907, 49201, 49060, 47948, 47724, 46990, 46544,
    46011, 45269, 44792, 44332, 43878, 43984, 42968, 42365, 42391, 42219, 41668, 41072, 40616, 40587,
    39999, 40169, 39340, 38906, 38438, 38142, 37757, 37818, 37535, 37217, 36757, 36589, 36151, 35953,
    35531, 35496, 35089, 35053, 34567, 34789, 34009, 33952, 33753, 33656, 33227, 32954, 32686, 32880,
    32709, 31886, 32126, 31657, 31466, 31142, 31106, 30650, 30316, 30494, 30328, 30157, 29611, 29754,
    29445, 28921, 29271, 29078, 28934, 28764, 28445, 28319, 28141, 28282, 27779, 27522, 27333, 27470,
    27289, 27102, 27018, 27066, 26925, 26384, 26188, 26385, 26392, 26082, 26062, 25660, 25682, 25547,
    25425, 25072, 25079, 25346, 24659, 24702, 24862, 24479, 24288, 24127, 24268, 24097, 23798, 23878,
    23893, 23817, 23398, 23382, 23280, 22993, 23018, 23242, 22987, 22894, 22470, 22612, 22452, 21996,
    21843, 22094, 21916, 21756, 21955, 21444, 21436, 21484, 21528, 21597, 21301, 21197, 21281, 21066,
    20933, 21023, 20888, 20575, 20574, 20511, 20419, 20312, 20174, 20023, 20087, 19955, 19946, 19846,
    19562, 19710, 19556, 19477, 19487, 19387, 19225, 19069, 19360, 18655, 19034, 18763, 18800, 19012,
    18893, 18714, 18645, 18577, 18317, 18458, 18374, 18152, 17822, 18102, 17735, 17940, 17805, 17711,
    17690, 17703, 17669, 17410, 17583, 17331, 17313, 16892, 16967, 16870, 16926, 17233, 16845, 16861,
    16576, 16685, 16455, 16687, 16747, 16524, 16473, 16349, 16273, 16255, 16228, 16219, 16021, 16111,
    15867, 15751, 16081, 15703, 15751, 15854, 15665, 15469, 15431, 15428, 15464, 15517, 15335, 15461,
    15237, 15292, 15305, 15351, 15078, 14810, 15119, 14780, 14664, 14869, 14722, 14890, 14672, 14439,
    14685, 14706, 14840, 14373, 14286, 14596, 14615, 14168, 14299, 13987, 14167, 14107, 14096, 14202,
    13985, 14118, 14094, 14127, 13896, 13864, 13597, 13572, 13717, 13669, 13782, 13617, 13284, 13333,
    13425, 13457, 13256, 13404, 13318, 13425, 13317, 13179, 13193, 13257, 13160, 12813, 13149, 13010,
    12867, 12958, 12818, 12801, 12749, 12810, 12575, 12673, 12514, 12735, 12523, 12677, 12298, 12469,
    12341, 12445, 12477, 12326, 12110, 12087, 12305, 12156, 12032, 12190, 12150, 11980, 12022, 11825,
    11969, 11831, 11997, 11924, 11739, 11685, 11702, 11783, 11783, 11659, 11647, 11610, 11526, 11577,
    11538, 11536, 11497, 11480, 11374, 11234, 11433, 11466, 11475, 11147, 11376, 11217, 11002, 11245,
    11124, 11000, 11129, 10923, 10966, 11071, 11029, 10972, 11012, 10800, 10936, 10904, 10750,
    10669, 10766, 10780, 10675, 10905, 10511, 10598, 10583, 10658, 10471, 10667, 10601, 10430, 10440,
    10510, 10148, 10468, 10346, 10257, 10286, 10235, 10351, 10182, 10182, 10095, 10192, 9866, 10070,
    10148, 9956, 10132, 10043, 9741, 10003, 10056, 9920, 10021, 9838, 9854, 9740, 9782, 9799,
    9798, 9788, 9840, 9747, 9797, 9893, 9593, 9535, 9658, 9554, 9593, 9530, 9523, 9488,
    9548, 9418, 9418, 9508, 9638, 9521, 9277, 9289, 9255, 9322, 9281, 9351, 9259, 9255,
    9225, 9098, 9268, 9227, 9224, 9106, 9239, 3815044], dtype=np.int64)

wikipedia_max_sequence_length = 512

squad_1_1_histogram = np.array([
    0, 0, 0, 0, 0, 0, 0, 0, 0, 0, 0, 0, 0, 0, 0, 0, 0, 0, 0, 0, 0, 0, 0, 0, 0, 0, 0, 0, 0, 0, 0, 0, 0,
    0, 0, 3, 2, 0, 9, 10, 16, 22, 24, 36, 35, 46, 42, 48, 57, 86, 83, 86, 87, 86, 97, 90, 99, 85, 94,
    105, 114, 110, 93, 116, 118, 114, 116, 117, 127, 115, 155, 137, 145, 157, 151, 153, 149, 163, 157,
    134, 150, 144, 132, 166, 162, 177, 160, 149, 151, 138, 156, 148, 176, 163, 182, 188, 182, 177, 199,
    182, 203, 201, 264, 250, 244, 289, 346, 327, 298, 377, 386, 444, 431, 503, 553, 532, 570, 611, 677,
    648, 673, 712, 722, 745, 692, 697, 747, 754, 741, 777, 781, 825, 813, 836, 777, 776, 756, 789, 790,
    765, 753, 729, 748, 772, 766, 760, 741, 725, 729, 759, 732, 730, 730, 741, 705, 708, 725, 656, 688,
    688, 677, 662, 628, 635, 618, 586, 527, 562, 619, 562, 578, 538, 558, 582, 541, 575, 526, 556, 498,
    529, 486, 528, 541, 482, 521, 483, 466, 514, 459, 447, 436, 383, 401, 408, 381, 369, 364, 381, 420,
    391, 388, 358, 365, 357, 358, 355, 297, 290, 267, 308, 329, 304, 332, 289, 282, 304, 242, 263, 288,
    238, 257, 271, 288, 277, 264, 253, 239, 217, 260, 214, 247, 237, 212, 205, 193, 200, 208, 195, 193,
    201, 187, 170, 176, 195, 156, 201, 179, 159, 183, 169, 178, 163, 153, 171, 144, 138, 181, 165, 171,
    161, 159, 166, 142, 138, 151, 155, 134, 141, 132, 123, 119, 109, 125, 123, 131, 135, 115, 108, 102,
    117, 105, 99, 84, 100, 85, 85, 85, 95, 122, 105, 114, 113, 100, 80, 96, 86, 79, 80, 87, 92, 73, 73,
    64, 76, 72, 77, 67, 60, 71, 77, 79, 72, 55, 67, 42, 59, 65, 72, 49, 43, 62, 48, 50, 54, 45, 42, 53,
    56, 45, 43, 32, 30, 36, 42, 37, 45, 28, 41, 31, 44, 35, 36, 47, 47, 48, 65, 32, 23, 35, 38, 20, 23,
    22, 21, 27, 20, 26, 18, 18, 22, 17, 17, 14, 26, 15, 20, 22, 19, 24, 17, 15, 20, 20, 22, 22, 17, 20,
    16, 21, 16, 23, 12, 14, 1054], dtype=np.int64)

squad_1)1_max_sequence_length = 384
```

Listing 7: Histogram creation for GLUE training datasets

```python
from transformers import AutoTokenizer
import datasets
import numpy as np

# constants
max_sequence_length = 128
task_to_keys = {
    "cola": ("sentence", None),
    "mnli": ("premise", "hypothesis"),
    "mrpc": ("sentence1", "sentence2"),
    "qnli": ("question", "sentence"),
    "qqp": ("question1", "question2"),
    "rte": ("sentence1", "sentence2"),
    "sst2": ("sentence", None),
    "stsb": ("sentence1", "sentence2"),
    "wnli": ("sentence1", "sentence2"),
}
glue_keys = ['cola', 'sst2', 'mrpc', 'qqp', 'stsb', 'mnli', 'rte', 'wnli']
# unused datasets due to missing training data
unglue_keys = ['mnli_matched', 'mnli_mismatched', 'qnli', 'ax']

# load data
dataset_loads = {}
for key in glue_keys:
    dataset_loads[key] = datasets.load_dataset("glue", key, split='train')

# tokenize data
tokenizer = AutoTokenizer.from_pretrained('bert-base-uncased')
tokenized_data = {}
for key in dataset_loads:
    sentence1_key, sentence2_key = task_to_keys[key]

    def preprocess_function(examples):
        """Tokenize the texts"""
        args = (
            (examples[sentence1_key],) if sentence2_key is None
            else (examples[sentence1_key], examples[sentence2_key])
        )
        result = tokenizer(*args, padding=False, max_length=max_sequence_length, truncation=True)
        return result

    tokenized_data[key] = dataset_loads[key].map(preprocess_function, batched=True)

# extract length information (for histogram plots)
histogram_length = {}
for key in tokenized_data:
    histogram_length[key] = []
for number, key in enumerate(tokenized_data.keys()):
    for raw_record in tokenized_data[key]["input_ids"]:
        histogram_length[key].append(len([x for x in raw_record if x!=0]))

# create histogram for packing
glue_histogram = {}
for data_key in histogram_length:
    glue_histogram[data_key] = np.array([0] * max_sequence_length, dtype=np.int64)
    for entry in histogram_length[data_key]:
        glue_histogram[data_key][entry-1] += 1
```

Listing 8: Longest-pack-first histogram-packing

```python
from collections import defaultdict
import numpy as np
import time

def add_pack(pack, count, tmp, final, limit, offset, max_sequence_length=512):
    """Filter out packs that reached maximum length or number of components."""
    # sanity checks
    assert(max_sequence_length-sum(pack) == offset), "Incorrect offset."
    assert(offset >= 0), "Too small offset."
    assert(offset < max_sequence_length), "Too large offset."
    if len(pack) == limit or offset == 0:
        final[offset].append((count, pack))
    else:
        tmp[offset].append((count, pack))

def pack_using_lpfhp(histogram, max_sequence_length, max_sequences_per_pack, distribute=True):
    """Longest-pack-first histogram-packing."""
    start = time.time()
    reversed_histogram = np.flip(histogram)
    # Initialize main strategy data dictionary.
    # The key indicates how many tokens are left for full length.
    # The value is a list of tuples, consisting of counts and respective packs.
    # A pack is a (sorted) list of sequence length values that get concatenated.
    tmp_strategies_per_length = defaultdict(list)
    strategies_per_length = defaultdict(list)
    if max_sequences_per_pack is "max":
        max_sequences_per_pack = max_sequence_length
    # Index i indicates here, how much space is left, due to reversed histogram
    for i in range(max_sequence_length):
        n_sequences_to_bin = reversed_histogram[i]
        length_to_bin = max_sequence_length - i
        offset = 0  # smallest possible offset for perfect fit
        while n_sequences_to_bin > 0:
            if (length_to_bin + offset) in tmp_strategies_per_length:
                # extract worst pack that will get modified
                n_sequences_to_pack, pack = tmp_strategies_per_length[
                    length_to_bin + offset].pop()
                # calculate how often the current sequence maximally fits in
                repeat = min(1 + offset // length_to_bin, max_sequences_per_pack-len(pack))
                # correct dependent on count
                while n_sequences_to_bin//repeat == 0:
                    repeat -= 1
                if not distribute:
                    repeat = 1
                new_pack = pack + [length_to_bin]*repeat
                count = min(n_sequences_to_pack, n_sequences_to_bin//repeat)
                if n_sequences_to_pack > count:
                    # old pack gets reduced
                    n_sequences_to_pack -= count
                    tmp_strategies_per_length[length_to_bin + offset].append(
                        (n_sequences_to_pack, pack))
                    n_sequences_to_bin -= count * repeat
                else:
                    n_sequences_to_bin -= n_sequences_to_pack * repeat
                add_pack(new_pack, count,
                         tmp_strategies_per_length, strategies_per_length,
                         max_sequences_per_pack, offset - (repeat - 1) * length_to_bin,
                         max_sequence_length)
                # clean up to speed up main key search
                if not tmp_strategies_per_length[length_to_bin + offset]:
                    tmp_strategies_per_length.pop(length_to_bin + offset)
                # reset offset in case best fit changed
                offset = 0
            else:
                offset += 1
            # Does not fit anywhere. Create new pack.
            if offset >= max_sequence_length - length_to_bin + 1:
                # similar repetition but no dependence on pack.
                repeat = min(max_sequence_length//length_to_bin, max_sequences_per_pack)
                while n_sequences_to_bin//repeat == 0:
                    repeat -= 1
                if not distribute:
                    repeat = 1
                add_pack([length_to_bin]*repeat, n_sequences_to_bin//repeat,
                         tmp_strategies_per_length, strategies_per_length,
                         max_sequences_per_pack, max_sequence_length-length_to_bin*repeat,
                         max_sequence_length)
                n_sequences_to_bin -= n_sequences_to_bin//repeat * repeat
```

```python
# merge all strategies
for key in tmp_strategies_per_length:
    strategies_per_length[key].extend(tmp_strategies_per_length[key])
# flatten strategies dictionary
strategy_set = []
strategy_repeat_count = []
for key in strategies_per_length:
    for count, pack in strategies_per_length[key]:
        pack.reverse()
        strategy_set.append(pack)
        strategy_repeat_count.append(count)

# Summarize efficiency of solution
duration = time.time() - start
sequence_lengths = np.arange(1, max_sequence_length + 1)
strategy_repeat_count = np.array(strategy_repeat_count)
n_strategies = len(strategy_set)
old_number_of_samples = histogram.sum()
new_number_of_samples = strategy_repeat_count.sum()
sequences = sum([count*len(pack) for count, pack in
                zip(strategy_repeat_count, strategy_set)])
total_tokens = max_sequence_length * new_number_of_samples
empty_tokens = sum([count*(max_sequence_length-sum(pack)) for count, pack
                   in zip(strategy_repeat_count, strategy_set)])
efficiency = 100 - empty_tokens / total_tokens * 100
speedup_upper_bound = 1.0/(1 - (histogram*(
    1 - sequence_lengths / max_sequence_length)).sum() / old_number_of_samples)

print(f"Packing efficiency (fraction of real tokens): {efficiency:3.4f}\n",
      f"Speed-up theoretical limit: {speedup_upper_bound:3.4f}\n",
      f"Achieved speed-up over un-packed dataset: {old_number_of_samples/new_number_of_samples:3.5f}",
      f"Runtime: Packed {old_number_of_samples} sequences in {duration:3.3f} seconds.")

return strategy_set, strategy_repeat_count
```

Listing 9: Extended non-negative least squares histogram-packing

```python
1  import time
2  import numpy as np
3  from scipy import optimize, stats
4  from functools import lru_cache
5
6  def get_packing_matrix(strategy_set, max_sequence_length):
7      num_strategies = len(strategy_set)
8      A = np.zeros((max_sequence_length, num_strategies), dtype=np.int32)
9      for i, strategy in enumerate(strategy_set):
10         for seq_len in strategy:
11             A[seq_len - 1, i] += 1
12     return A
13
14 @lru_cache(maxsize=None)
15 def get_packing_strategies(start_length, minimum_increment, target_length, depth):
16     gap = target_length - start_length
17     strategies = []
18     # Complete the packing with exactly 1 number
19     if depth == 1:
20         if gap >= minimum_increment:
21             strategies.append([gap])
22     # Complete the sample in "depth" steps, recursively
23     else:
24         for new in range(minimum_increment, gap + 1):
25             new_gap = target_length - start_length - new
26             if new_gap == 0:
27                 strategies.append([new])
28             else:
29                 options = get_packing_strategies(start_length + new, new, target_length, depth - 1)
30                 for option in options:
31                     if len(option) > 0:
32                         strategies.append([new] + option)
33     return strategies
34
35 def pack_using_ennlshp(histogram, max_sequence_length, max_sequences_per_pack):
36     # List all unique ways of packing to the desired maximum sequence length
37     strategy_set = get_packing_strategies(0, 1, max_sequence_length, max_sequences_per_pack)
38     print(f"Packing will involve {len(strategy_set)} unique packing strategies.")
39     # Get the packing matrix corresponding to this list of packing strategies
40     A = get_packing_matrix(strategy_set, max_sequence_length)
41     # Weights that penalize the residual by the number of resulting padding tokens.
42     w0 = np.array([x+1 for x in range(max_sequence_length)])
43     # construct the packing matrix
44     A_bar = np.zeros((2*max_sequence_length, len(strategy_set) + max_sequence_length), 'd')
45     # Base weighted matrix
46     A_bar[:max_sequence_length, :len(strategy_set)] = np.expand_dims(w0, -1) * A
47     # Higher weight to avoid positive residual
48     A_bar[max_sequence_length:, :len(strategy_set)] = np.expand_dims(
49         10**6*np.ones([max_sequence_length]), -1) * A
50     # negative diagonal unity matrix for mapping to residual
51     A_bar[max_sequence_length:, len(strategy_set):] = np.expand_dims(
52         10**6*np.ones([max_sequence_length]), -1)*np.ones((max_sequence_length,max_sequence_length))
53     b_bar = np.zeros(2*max_sequence_length)
54     # Apply weighting to histogram vector
55     b_bar[:max_sequence_length] = w0 * histogram
56     b_bar[max_sequence_length:] = 10**6*np.ones([max_sequence_length]) * histogram
57     # Solve the packing problem
58     print(f"Sequences to pack: ", histogram.sum())
59     start = time.time()
60     strategy_residual, rnorm = optimize.nnls(A_bar, b_bar)
61     strategy_repeat_count = strategy_residual[:len(strategy_set)]
62     print(f"Solving non-negative least squares took {time.time() - start:3.2f} seconds.")
63     # Round the floating point solution to nearest integer
64     strategy_repeat_count = np.rint(strategy_repeat_count).astype(np.int64)
65     # Compute the residuals, shape: [max_sequence_length]
66     residual = histogram - A @ strategy_repeat_count
67     # Handle the left-over sequences i.e. positive part of residual
68     unpacked_seqlen = np.arange(1, max_sequence_length + 1)[residual > 0]
69     for l in unpacked_seqlen:
70         strategy = sorted([l, max_sequence_length - l])  # the depth 1 strategy
71         strategy_index = strategy_set.index(strategy)
72         strategy_repeat_count[strategy_index] += residual[l-1]
73     # Re-compute the residual with the updated strategy_repeat_count
74     # This should now be strictly < 0
75     residual = histogram - A @ strategy_repeat_count
76     # Add padding based on deficit (negative residual portion of residual)
77     padding = np.where(residual < 0, -residual, 0)
78     # Calculate some basic statistics
79     sequence_lengths = np.arange(1, max_sequence_length + 1)
80     old_number_of_samples = histogram.sum()
81     new_number_of_samples = int(strategy_repeat_count.sum())
82     speedup_upper_bound = 1.0/(1 - (histogram*(
83         1 - sequence_lengths / max_sequence_length)).sum()/old_number_of_samples)
84     num_padding_tokens_packed = (sequence_lengths * padding).sum()
85     efficiency = 1 - num_padding_tokens_packed/(new_number_of_samples*max_sequence_length)
86     print(f"Packing efficiency (fraction of real tokens): {efficiency:3.4f}\n",
87           f"Speed-up theoretical limit: {speedup_upper_bound:3.4f}\n",
88           f"Achieved speed-up over un-packed dataset: {old_number_of_samples/new_number_of_samples:3.5f}")
89     return strategy_set, strategy_repeat_count
```

APPENDIX REFERENCES

G. Belov and G. Scheithauer. A branch-and-cut-and-price algorithm for one-dimensional stock cutting and two-dimensional two-stage cutting. *European Journal of Operational Research*, 171 (1):85–106, may 2006. ISSN 03772217. doi: 10.1016/j.ejor.2004.08.036. URL https://linkinghub.elsevier.com/retrieve/pii/S0377221704006150.

Michael R. Garey and David S. Johnson. *Computers and Intractability; A Guide to the Theory of NP-Completeness*. W. H. Freeman & Co., USA, 1990. ISBN 0716710455.

Gabriel Guillén, Claudia Diaz-Camino, Carlos Loyola-Torres, Rosaura Aparicio-Fabre, Alejandrina Hernández-López, Mauricio Díaz-Sánchez, and Federico Sanchez. Detailed analysis of putative genes encoding small proteins in legume genomes. *Frontiers in Plant Science*, 4:208, 2013. ISSN 1664-462X. doi: 10.3389/fpls.2013.00208. URL https://www.frontiersin.org/article/10.3389/fpls.2013.00208.

Henrik B. Hansen, Peter B. Damgaard, Ashot Margaryan, Jesper Stenderup, Niels Lynnerup, Eske Willerslev, and Morten E. Allentoft. Comparing ancient dna preservation in petrous bone and tooth cementum. *PLOS ONE*, 12(1):1–18, 01 2017. doi: 10.1371/journal.pone.0170940. URL https://doi.org/10.1371/journal.pone.0170940.

S. Kotz and S. Nadarajah. *Extreme Value Distributions*. World Scientific Publishing Company, 2000. ISBN 9781783261734.

Charles L. Lawson and Richard J. Hanson. *Solving Least Squares Problems*. Society for Industrial and Applied Mathematics, jan 1995. ISBN 978-0-89871-356-5. doi: 10.1137/1.9781611971217. URL http://epubs.siam.org/doi/book/10.1137/1.9781611971217.

C. C. Lee and D. T. Lee. A Simple On-Line Bin-Packing Algorithm. *Journal of the ACM (JACM)*, 32(3):562–572, jul 1985. ISSN 1557735X. doi: 10.1145/3828.3833. URL https://dl.acm.org/doi/10.1145/3828.3833.

Y. Luo and Ramani Duraiswami. Efficient parallel non-negative least squares on multi-core architectures. *SIAM Journal on Scientific Computing*, 33:2848 – 2863, 2011.

NVIDIA. Performance catalogue for BERT on Pytorch. https://ngc.nvidia.com/catalog/resources/nvidia:bert_for_pytorch/performance, 2021.

Yifan Peng, Shankai Yan, and Zhiyong Lu. Transfer Learning in Biomedical Natural Language Processing: An Evaluation of BERT and ELMo on Ten Benchmarking Datasets. In *Proceedings of the 2019 Workshop on Biomedical Natural Language Processing (BioNLP 2019)*, pp. 58–65, 2019.

Pauli Virtanen, Ralf Gommers, Travis E. Oliphant, Matt Haberland, Tyler Reddy, David Cournapeau, Evgeni Burovski, Pearu Peterson, Warren Weckesser, Jonathan Bright, Stéfan J. van der Walt, Matthew Brett, Joshua Wilson, K. Jarrod Millman, Nikolay Mayorov, Andrew R. J. Nelson, Eric Jones, Robert Kern, Eric Larson, C J Carey, İlhan Polat, Yu Feng, Eric W. Moore, Jake VanderPlas, Denis Laxalde, Josef Perktold, Robert Cimrman, Ian Henriksen, E. A. Quintero, Charles R. Harris, Anne M. Archibald, Antônio H. Ribeiro, Fabian Pedregosa, Paul van Mulbregt, and SciPy 1.0 Contributors. SciPy 1.0: Fundamental Algorithms for Scientific Computing in Python. *Nature Methods*, 17:261–272, 2020. doi: 10.1038/s41592-019-0686-2.

Thomas Wolf, Quentin Lhoest, Patrick von Platen, Yacine Jernite, Mariama Drame, Julien Plu, Julien Chaumond, Clement Delangue, Clara Ma, Abhishek Thakur, Suraj Patil, Joe Davison, Teven Le Scao, Victor Sanh, Canwen Xu, Nicolas Patry, Angie McMillan-Major, Simon Brandeis, Sylvain Gugger, François Lagunas, Lysandre Debut, Morgan Funtowicz, Anthony Moi, Sasha Rush, Philipp Schmidd, Pierric Cistac, Victor Muštar, Jeff Boudier, and Anna Tordjmann. Datasets. *GitHub. Note: https://github.com/huggingface/datasets*, 1, 2020.

Wolfram Research Inc. Mathematica, Version 12.2. URL https://www.wolfram.com/wolfram-alpha-notebook-edition. Champaign, IL, 2020.

