# OpenReview forum: "Efficient Packing: Towards 2x NLP Speed-Up without Loss of Accuracy for BERT"
_ICLR.cc/2022/Conference — ICLR 2022 Submitted_

### Official Review · Reviewer_nPLY · 2021-10-31

**Correctness:** 3
**Technical Novelty And Significance:** 3
**Empirical Novelty And Significance:** 4
**Recommendation:** 5
**Confidence:** 4

**Main Review:**

Most part of the paper is clearly presented. The proposed packing strategy should be novel, easy to implement and also effective. However, I have the following concerns and comments.

1. The paper claimed the packing algorithm doesn't loss accuracy compared with the original algorithm by showing the training loss and training accuracy. However, since the sequences are not i.i.d. in the packing algorithm, I am wondering if the generalization performance on downstream tasks (such as GLUE) will be affected.
2. As mentioned in the paper, BERT is pre-trained in two phases, where the first phase uses sequence length 128 for 900K steps and the
second phase uses sequence length 512 for 100K steps. Therefore, if compared with this approach, the speedup from the proposed method is probably only 1.2x instead of 2x.
3. Why do we need a limit for pack depth?
4. Other than the original training approach, the paper only compares with the un-packing methods in multi-accelerator case. It lacks comparison with baselines such as grouping samples by size before batching as mentioned in Page 2.

Minor:
1. `s_m` in Sec. 3 is not clearly defined.
2. The abstract mentioned 0.02 seconds overhead time for SPFHP and 28.4 seconds for NNLSHP. In the main text, I only see 0.03 seconds for SPFHP and no mentioning about NNLSHP.

**Summary Of The Paper:**

The paper proposed to pack sequences instead of padding to reach the max sequence length for each sample for BERT pretraining. To achieve similar optimization results with the original training process, the proposed method modified positional embedding, additional attention masks and other optimization hyper-parameters. On the Wikipedia dataset, the proposed method can achieve 2x speedup while achieving similar training loss.


**Summary Of The Review:**

Overall, the paper proposed a simple and effective method for speeding up BERT training especially on the datasets with a large variance of sequence length. However, due to lack of comparison with other similar baseline methods, I think the current version is slightly below the acceptance threshold.

---

> ### Comment · Reviewer_nPLY · 2021-11-23
> **Have read the comments and keep my score**
>
> I have read authors' comments. But I don't think the authors have addressed my 4th concern, comparison with baselines such as grouping samples. I'll keep my score.

---

> > ### Author Response · Authors · 2021-11-26
> > **Discussion of Baselines**
> >
> > We thank the reviewer for the updated feedback and we are glad that most concerns/comments were answered.
> >
> > In the revised version of the paper, we added the sorted batching (SORT) as well as GREEDY packing, as for example applied for the T5 training dataset generation, to address other packing approaches.
> >
> > From a packing efficiency point of view, SORT is the most efficient algorithm with more than 99.99% efficiency. However, SORT comes with a lot of drawbacks which in our case result in performance worse than naive batching. Hence, we used naive batching instead as a baseline, since it shows better performance. There are multiple arguments that speak against SORT which are highlighted in the paper in the third paragraph in the introduction.
> >
> > 1. SORT requires a lot of optimizations depending on the hardware and is not hardware independent, which is what we focus on in our paper. In the field of hardware specific solutions, un-padding has shown the better performance in the MLPerf(TM) competition than SORT. Thus, it is the better baseline.
> > 2. With each different sequence length, a different batch size is required for load balancing. Otherwise, the hardware is not sufficiently used. This requires fine tuning for each hardware and sequence length.
> > 3. With the dynamically changing batch sizes however, the learning rate needs to be readjusted for each sequence length. This is intractable. Thus, convergence behavior is impacted. This is another reason, why un-padding is currently more established for hardware acceleration to get the fastest time to train (see MLPerf(TM) submissions).
> > 4. Each different sequence length and batch size in this case results in a different computational graph. Optimizing the graphs is crucial for almost any hardware as explained in our provided XLA references in the paper. Each graph needs a different distribution of the workloads and thus different computational kernels or code layouts/distributions for optimal performance. Hence, we have not one setup but more than one hundred setups, that get continuously changed. This can impact usability and performance depending on software and hardware setup.
> >
> > Any suggestions or feedback about these points and how to address them better in the paper would be greatly appreciated.

---

### Official Review · Reviewer_EN5A · 2021-10-31

**Correctness:** 4
**Technical Novelty And Significance:** 3
**Empirical Novelty And Significance:** 2
**Recommendation:** 6
**Confidence:** 3

**Main Review:**

Strongness:
(1) The method is easy to understand and easy to incorporate into the existing training pipeline.
(2) The performance gain looks great.
(3) Have the ablation study to show the effectiveness of the model arch changes.
(4) Easy to understand

Weakness:
(1) Although pre-training is an important part of BERT, did you test the different pre-trained models on downstream tasks?
(2) I did not find the GitHub link in Appendix. The text code is hard to do the reproduction.

Questions:
(1) In section 3.3, the authors mentioned the learning rate is not adjusted since the gradient is accumulated by averaging. However, as far as I know, LAMB also uses the averaging of the gradient but it still needs to scale the learning rate. Can you explain more?
(2) Do you have a comparison of the two proposed packing algorithms?

**Summary Of The Paper:**

The paper proposes a simple way to pack sentences together for efficient training. In order to recover the accuracy/performance, several model arch and optimizer hyper-parameter changes are introduced.

**Summary Of The Review:**

Please see above.

---

### Official Review · Reviewer_xs2k · 2021-11-03

**Correctness:** 3
**Technical Novelty And Significance:** 2
**Empirical Novelty And Significance:** 2
**Recommendation:** 5
**Confidence:** 4

**Main Review:**

The motivation makes great sense and the overall structure of the paper is very clear (though I did not read anything in the appendix, i.e., from page 13-37).

Usually dynamic sequence length is not supported in GPU or TPU training libraries. The most common way is to pad a fixed sequence length (e.g., 512) and provide a mask in the input so the padded data can be masked out during computation. The advantage of this approach is that this is straightforward to implement.  However, it loses efficiency by padding all the data into the max sequence length.

The proposed work is to reduce padding via packing multiple sequences into one fixed length. I am not familiar with publications on this front, but I am aware of an existing implementation of similar methodology in `tensor2tensor` library, there is a `pack_dataset` method (https://github.com/tensorflow/tensor2tensor/blob/master/tensor2tensor/data_generators/generator_utils.py#L671), which is designed following a very similar idea. This implementation in tensor2tensor is not profiling the dataset on sequence length histogram because it is receiving the data in a streaming fashion. So I agree that this paper is definitely doing something related but differently by considering the entire problem as bin-packing so as to look for approximate optimal global solution.

General Significance: The main significance is the two packing algorithms being proposed and experimented in BERT pre-training. Multiple sequences can be packed into one fixed length. However, I see the impact is limited because:
1) this strategy is data dependent. Though there could be many datasets that have this property, there are also other datasets that could follow a different sequence length distribution in which case the proposed methodology won't achieve similar amount of speedup.
2) because the packing algorithm enforces a specific order (by selecting which sequence to add to which pack), this affects the i.i.d assumption. Also this assumption is actually important in many data preprocessing pipelines we use for BERT encoder. If we have to use one model (packed) for training pre-trained BERT model, then using another model (non-packed dataset and modeling code) for training downstream models this is really not ideal in development.
3) this workaround (to reduce padding) creates complexity in the user code, both in the input pipelines and the modeling changes. And more importantly, as the author already pointed out, this workaround also increases the effective batch size which might need a complete re-tuning of hyper-parameters of the model.

Novelty: A baseline implementation of multi-sequence-packing already exists in a public library (tensor2tensor). The author might want to clarify what are the extensions on top of tensor2tensor. As I pointed above, tensor2tensor is consuming data in a streaming fashion, and  this paper is considering the entire problem as bin-packing so as to look for approximate optimal global solution. Could you quantify the difference between using tensor2tensor library vs your proposed two bin-packing algorithms regarding speedup vs quality?

Technical quality: The overall technical quality is limited because of the limitation mentioned in "significance" and "novelty".

Clarity: The paper is well-structured, with clear relationship between each section. The ideas are clearly conveyed with a good writeup.

Minor issue(s):
* Formatting problem in section 2 before and after CLS and SEP token. (¡CLS¿, ¡SEP¿ , which I guess they should be <.>) Hopefully this is not a problem from my browser viewing pdf file.

**Summary Of The Paper:**

This paper profiles the training data of BERT and finds out opportunities to reduce padding thus saving computation. By packing multiple sequences into one fixed length sequence (two packing algorithms were used), the author shows that a BERT training speed-up of 2x can be achieved without loss of quality. The author also studies how to correspondingly change the modeling configuration and optimizer configurations to compensate the effective batch size increase due to the packing.

**Summary Of The Review:**

marginally below the acceptance threshold

---

> ### Comment · Reviewer_xs2k · 2021-11-14
> **Have read other reviewer's coments**
>
> I am still keeping my ratings unchanged. The author needs to answer the question we ask.

---

> ### Comment · Reviewer_xs2k · 2021-11-16
> **Have read the author's response**
>
> Thanks for the response from the author(s).
> I am still holding my rating. The limitations of the paper still hold. My main concern is that given the limitation and the pre-existing work from T5 (tensor2tensor library) the contribution from the paper is below the bar of ICLR.
> I do agree that both the idea and the tooling the author is going to share would be helpful for the community, but the paper itself does not meet the bar of significance and novelty.

---

> > ### Author Response · Authors · 2021-11-18
> > **Performance of greedy packing as in T5**
> >
> > T5 requires the C4 dataset, which we have not currently available. However, to give an idea of the difference in packing efficiency and acceleration, we can analyse the performance of greedy aggregation of samples on our given Wikipedia dataset.
> >
> > We take the histogram and cast it back to a list of different sequence lengths since this is all that matters for analysing packing behaviour. Next, we randomly shuffle the dataset and iterate with the greedy aggregation algorithm multiple times to account for randomness. We iterate sequence by sequence and combine them provided the maximum sequence length of 512 is not reached. If it is exceeded, the packed sequence is considered finished and a new sequence is started.
> >
> > The greedy packing algorithm itself takes a bit more than 10 seconds, since we are operating on single sequences and not histogram counts. The efficiency of this approach is 78.24% (standard deviation of 0.005) compared to our 99.75% for NNLSHP. The respective acceleration would be around 1.566x compared to our 2x. With respective separator tokens, the performance decreases around 0.13% for one separator token and 0.27% when two separator tokens are required between two sequences. Following the brief documentation at https://github.com/tensorflow/tensor2tensor/blob/5623deb79cfcd28f8f8c5463b58b5bd76a81fd0d/tensor2tensor/data_generators/generator_utils.py#L1086, two separator tokens would be expected.
> >
> > In addition to the packing preprocessing, our paper proposes, rather than using separator tokens, to instead modify the masking of the attention matrix during training. The RoBERTa paper shows that avoiding contamination of sequences from different documents can consistently improve downstream F1 scores by 0.35%.
> >
> > Would it be sufficient to add a respective section to the appendix? If it should be part of the main paper, which part should it replace?

---

### Official Review · Reviewer_g5tJ · 2021-11-03

**Correctness:** 3
**Technical Novelty And Significance:** 3
**Empirical Novelty And Significance:** 2
**Recommendation:** 6
**Confidence:** 3

**Main Review:**

Generally, simple packing ideas are frequently used such as sorting sequences by their lengths and then try to “cluster” near length sequences into one mini-batch. This paper further goes deeper into two interesting packing algorithms and experimented on IPU hardware with Wikipedia style datasets and BERT-large models.

Detailed questions and comments:
1.	Possibly trying larger models such as GPT3 and later will be an important directly to testify your ideas: bert-large is not that big enough and saving 50% training time will be more meaningful for super-large models such as GPT3.
2.	Wikipedia is not generally used for domain specific bert such as healthcare and finance domains, in which pubmed and financial news are frequently used. So, the findings of specific datasets are quite limited and hard to say it is a general finding.
3.	Not find any baseline packing algorithms here. Any simple heuristics used in the baselines? Such as sorting by length and then perform the training with as fewer packing symbols as possible in minibatch.
4.	From reading the paper, I think the two algorithms are not limited to BERT (its MLM and NSP pretraining tasks) and really willing to see other types of pretraining tasks, such as T5, GPT style’s results. These will make an evaluation of these two algorithms’ novelty and generalization ability to be with richer evidences.
5.	Also wonder if the packing algorithms can be applied to ASR tasks such as packing audios as well. It looks that not only applicable to textual sequences.
6.	IPU is less available to most readers yet, so possibly also mentions details in GPU/TPU are preferred as well.


**Summary Of The Paper:**

This paper proposes two packing algorithms for bert pretraining, shortest-pack-first histogram-packing (SPFHP) and non-negative least-squares histogram-packing (NNLSHP). 2* speed was achieved under datasets such as Wikipedia for bert-large training.These packing algorithms packed Wikipedia’s 16M sequences in 0.02s (SPFHP). Packing depths of from 1 to 16 were testified. This paper also has 20 pages near appendix telling about packing algorithms, packedBERT of model changes and hypermeter adjusting, and detailed experiments such as bin-packing algorithm comparison, scaling analysis and technical background on packing with some core codes attached as well.

**Summary Of The Review:**


Strong:
1.	Two packing algorithms for bert-large and related pretraining models;
2.	2* speeding up without losing accuracy for bert-large pretraining under datasets such as Wikipedia;
3.	Rich analysis of technical backgrounds, experiments, and code details.
Weak:
1.	Limited to Wikipedia and bert-large is a small scope considering that there are rich pretraining tasks, rich domain and rich datasets of different languages;

---

> ### Comment · Reviewer_g5tJ · 2021-11-17
> **about point 6 IPU vs. GPU/TPU**
>
> thanks the authors' feedbacks, and for point 6,
>
> "It would be helpful if reviewer g5tJ could provide a clarification on point 6. As mentioned in the paper, our algorithm is hardware independent and can be transferred to GPU and TPU with different software frameworks like PyTorch and TensorFlow. In fact, we know that our approach has been applied in a different GPU application and for BERT it has been applied on new modern hardware beyond IPUs. Which additional details would be useful to provide in the paper?" -> noticed that your method is hardware independent, just wondering if you can report/share some comparison experiment results to show that under both IPU and GPU, the algorithms work well.
>
> "we know that our approach has been applied in a different GPU application" -> is it possible to give me the reference?

---

> > ### Author Response · Authors · 2021-11-17
> > **Hardware limitation?**
> >
> > We thank reviewer g5tJ for the clarification.
> >
> > At the current stage, we don't have the code, results, or respective references for other hardware and won't be able to generate them before the November 22nd deadline. For future revisions of the paper, we will add respective result and code references in TensorFlow and PyTorch.

---

> > ### Author Response · Authors · 2021-12-02
> > **Transfer to different hardware and TensorFlow**
> >
> > Today, the code for running our algorithm with TensorFlow on the Habana Gaudi chip has been released. The acceleration showed the expected improvement of around 2x.
> >
> > https://github.com/mlcommons/training_results_v1.1/tree/main/Intel-HabanaLabs/benchmarks/bert/implementations/TensorFlow/nlp/bert

---

> ### Author Response · Authors · 2021-11-22
> **Pubmed and financial news and generalization to other data**
>
> Our findings are more general and sequence length variability occurs everywhere. We found evidence for example in "in DNA sequence lengths (Hansen et al.,2017) and protein lengths (Guill ́en et al., 2013)" and showed respective results for SQuAD and some GLUE datasets in our paper. In everything text related like sentences, paragraphs, and full documents, a large variability in length can be observed. Only if a document has a clearly predefined structure and limited amount of content that can be put inside, resulting sequence lengths will show little variation. This could be for example holding for questionnaires and some medical documents. Abstracts, usually show a large variation in length (depending on the upper limit on length). Conference papers on the other side, only when considered without the references and appendix, will not show much variation in length. Only the amount and size of images will have a minor impact. On the other hand, with the appendix, page counts will vary between 6 and 30 pages with a clear peak around 8, which will make packing very beneficial for them if they get directly combined. For the LibriSpeech dataset, we discovered that the resulting audio sequences are rather long and packing can achieve only a 1.3x acceleration and 20% padding remain.
>
> Depending on the data generation scheme, sizes can vary, too. For example, if to long documents are cut short and the overhead data is discarded, packing will show no benefit. In contrast, BERT is cutting sequences randomly short by design and thus can cause the resulting length distribution to be even more skewed than what can be seen in the raw data.
>
> To provide respective results on pubmed and financial news, it would be great to get some guidance for the analysis in form of histograms or references that help to implement the respective data download and preprocessing. For pubmed, the standard BERT dataset creation with sequence length 512 is applied as described in the BlueBERT paper? Is financial news referring to FinBERT?

---

### Official Review · Reviewer_4Mr1 · 2021-11-08

**Correctness:** 3
**Technical Novelty And Significance:** 2
**Empirical Novelty And Significance:** 1
**Recommendation:** 3
**Confidence:** 4

**Main Review:**

Strength:

The idea is simple and easy to understand.


Weakness

1. The authors revealed their identity (affiliation) in the code snippets of appendix. It might violate the double-blind rule of ICLR.

2. The novelty of this paper is limited. Packing is not a new idea and it has been widely used in the official tensor2tensor library and achieved good results: https://github.com/tensorflow/tensor2tensor/blob/3f12173b19c1bad2a7c37eb390f3ad46baee0c19/tensor2tensor/data_generators/ops/pack_sequences_ops.cc. So this can be a useful trick but the contribution might be not significant enough to publish at ICLR.

3. The scenario discussed in this paper is too restricted. For example, 1) it only discuss the wikipedia dataset (that’s how the number 50% comes), but there are a lot more datasets; 2) it only works for BERT training, but there are quite a few other important tasks, such as language modeling (GPT-3). All the numbers reported in this paper are based on this setting, making its generalization capability questionable.

4. The experiment section only shows the training loss of pretraining, but never talks about the downstream fine-tuning. Then how do you conclude that the performance is little affected? After all, the accuracies of downstream applications is the final metric.

5. The paper is a bit hard to read for the following reasons: 1) The contents are not self-contained in the main text — quite a few important contents are deferred to appendix that one cannot easily follow the ideas in the main text; 2) The paragraphs are usually lengthy and verbose —  they can be as long as 30 lines! 3) There are quite a few typos, e.g. “For achieve this”, “¡CLS¿” etc.

**Summary Of The Paper:**

This paper proposes efficient packing methods for training sequences of BERT, such that the 50% of the padding tokens in the Wikipedia dataset is avoided to speed up the training. These methods include shortest-pack-first histogram-packing (SPFHP) and non-negative least-squares histogram-packing (NNLSHP) algorithms, which are shown to be straightforward to implement and have little impact on the performance. Empirical studies show that a near 2x speedup over the vanilla BERT training is achieved by the proposed methods.

**Summary Of The Review:**

The paper violates the double-blind rule (in the appendix). It lacks novelty, only works on a restricted setting and is a bit hard to read.

---

> ### Comment · Reviewer_4Mr1 · 2021-11-28
> **Score unchanged**
>
> Since the authors did not specifically address my concerns, I will keep my score unchanged. I still think the novelty is limited, the scenario is restricted and the experiments are not convincing as the downstream applications are not properly discussed in the main text. The double-blind violation is also a factor that contributes to my score.

---

> > ### Author Response · Authors · 2021-11-29
> > **Specific address of reviewer concerns**
> >
> > 1. The code samples provide license statements such that others can easily use the code. It does not provide any information about the authors of this paper.
> > 2. Packing is much older than tensor2tensor (and the respective T5 model) as outlined in our state of the art. The revised paper now also compares against the respective tensor2tensor, T5, greedy, packing approach. The contribution of our paper is not to introduce packing. The contributions, as highlighted in our main reply, are the literature overview and visualizations about packing, two new packing approaches that are much more efficient than anything provided in the literature so far especially in the sense of complexity, and most importantly a totally new approach to avoid cross contamination between samples and to make sure that performance does not go down with packing. The only paper that addressed the reduction in performance from packing, so far, is the RoBERTa paper but it does not provide a solution to it. The adjustments of losses, hyperparameters, positional encodings, and attention masks are the main contribution of our paper and applicable far beyond BERT or NLP as discussed in the paper.
> > 3. We now show in the paper that packing other datasets like SQuAD, GLUE, PubMed, LibriSpeech is beneficial, too, and not just Wikipedia. We address how our approach can be beneficial for other models like RoBERTa and T5, too. Both suffer from cross-contamination.
> > 4. We now provide a downstream result on SQuAD in the paper that confirms that predictive performance is not impacted. We also have an ablation study that confirms that our adjustments are necessary to achieve performance matching.
> > 5. The compression of the rich content from 40 down to 8 pages is a challenging topic especially in context of requested more datasets, more models, and required clarifications on the state of the art. Which important content from the appendix should be moved into the 8 pages? Which content should be removed or moved to the appendix? The review process has shown us that there are a lot of misconceptions in the domain of packing and a lot of aspects that require longer explanations. We corrected the two typos that were pointed out, had extra review of the added sections, and will have another professional proof reading for the final version.

---

### Comment · Area_Chair_DsC9 · 2021-11-14
**Additional Discussion Encouraged**

Dear Reviewers,

can you please take a look at each other's reviews? Your reviews currently straddle the decision boundary and it would be good to make sure you have considered all the perspectives provided. Please update your reviews (at least to acknowledge that you have read all reviews).

Thanks,
Your Area Chair

---

### Author Response · Authors · 2021-11-16
**Major reply: We thank the reviewers for their detailed feedback and are currently working through it.**

Most reviewers found the paper easy to read, which is reassuring. Given the current length of the paper, it is becoming increasingly tricky to add more content and explanations. Hence, any recommendations for rearrangement or shortening of the main paper part are appreciated.

Our paper clearly states that the concept of packing is nothing new and we thank the reviewers for pointing us to the T5 approach. The novelty of the paper comes from 4 aspects as highlighted partially by the reviewers:

- We provide the first literature overview of packing approaches or related algorithms in NLP. The T5 packing approach from the tensor2tensor library is like the RoBERTa and GPT3 approach on “packing”, according to [1]. Thus, this approach is not applicable to BERT, which has a different data generation approach. Furthermore, the arguments in our paper related to GPT3 and RoBERTa also hold for T5, especially that these models will not benefit from our packing acceleration since their sequences are already at almost full length. An interesting aspect of the T5 code is that it adds additional padding tokens between concatenated sequences. Hence, it would be interesting ablation study to determine how many padding tokens are required in between sequences to avoid cross-contamination.
- We are the first paper to visualize and address the impact of padding on wasted compute. We show that packing is important not only for the Wikipedia dataset but also other datasets like SQuAD and GLUE datasets. Whereas for Wikipedia, one could argue that the imbalance comes from the BERT dataset generator, the other datasets show that sequence length imbalance can be found in a wide range of applications. In the meantime, we also observed length differences in Graphs and audio sequences. Hence, we are confident that it is also present in healthcare and finance domains (as requested by reviewer g5tJ). This shows that our packing can be applied to ASR tasks, too, as mentioned by reviewer g5tJ.
- We provide 2 new packing algorithms with much better efficiency than the state of the art in terms of time and space complexity.
- We provide a novel approach to modify transformer-based models to avoid cross-contamination between packed sequences by modifying the attention mask and unpacking the loss. In contrast to existing approaches, we provide the first approach that is hardware independent and can be used for acceleration on any hardware accelerator. As now mentioned in our future work, we are curious about how far GPT3/RoBERTa/T5 would benefit from avoiding cross-contamination. Note, however, that our ablation study as well as the RoBERTa paper already proved that avoiding cross-contamination improves predictive performance in pretraining. Thus, we provide the first simple approach to obtain this benefit. We understand that the reviewers would like to see these results already in the current publication, which we will not yet be able to provide due to resources.

Reviewer xs2k correctly mentions that “Usually dynamic sequence length is not supported in GPU or TPU training libraries.” Thus, sorted batching (Faster Transformer) and Efficient transformer are hardware-specific solutions requiring special software handling and do not generalize well. Hence, we focus our comparison on the only hardware independent solution as a baseline, which is vanilla padding. If we provided results on our hardware with sorted batching, timings would be worse than with vanilla padding.

We understand that generalization performance of the model is a concern despite our approach closely matching loss and accuracy. We will provide updated results on the SQuAD downstream task to show that performance is not impacted (if we manage to get the results in time). Currently, we do not have the results.

We agree with the reviewers that the performance impact can vary from application to application. We try to address this in the appendix by looking at different datasets and by providing the histograms for Wikipedia with different maximum sequence lengths. We are open to any suggestions to better address this aspect. We include scripts to provide an easy means of determining packing efficiency for the application at hand. We disagree with the assessment of reviewer nPLY on the efficiency for BERT full pre-training. Phase 1 exists because of the large amount of padding for sequence length 512 and because computational complexity and memory required of the self-attention increases not linearly but squared. Hence, the time for processing 900k steps of length 128 is like processing 100k steps with sequence length 512? Thus, the overall benefit would be still 1.6x.

References:
[1] Exploring the Limits of Transfer Learning with a Unified Text-to-Text Transformer

---

> ### Author Response · Authors · 2021-11-18
> **Downstream performance**
>
> We will add the following section to the appendix and reference it in the main article:
>
> ```
> \section{SQuAD 1.1}
> \label{a:downstream}
>
> Packing slightly violates the i.i.d. assumption of data.
> Thus, it is of interest, if the algorithm matches downstream performance.
> This is especially relevant with a full training setup without a starting checkpoint.
> We trained Phase 1\&2 of BERT base with and without packing.
> To avoid giving an advantage to packing by further hyperparameter tuning,
> we instead reduced the gradient accumulation count for the packed BERT training
> for Phase 1 and Phase 2 to match the total number of sequences that get processed.
> With this approach, we could use the same hyperparameters and number of training steps.
> This gives a slight disadvantage to the packed run.
> For Phase 2, we used sequence length $348$
> since longer range attention is not relevant for SQuAD 1.1.
> For the fine-tuning training on SQuAD 1.1, we did not use packing.
> After $10$ repetitions, the results showed that on average,
> the F1 score is reduced by $0.003\%$ whereas the EM score is improving by $0.049\%$.
> ```
>
> We are aware that this is not a full downstream analysis.

---

### Author Response · Authors · 2021-11-16
**Minor comments**

Minor comments:

- We fixed the three typos concerning <CLS> and <SEP> token and “To achieve this”.
- We added the definition of s_m and n into the introduction of the methods section.
- We added the timing for NNLSP to the results and corrected the report times.
- We added the reference to [1] to the paper.
- The code samples are provided with links to be downloaded since copying from the paper is always cumbersome. The GitHub links for the core code samples are activated when clicking on the image (at reviewer EN5A). We also provide the raw files as supplemental material.
- We have limited the packing depth because, depending on the hardware and software implementation, it can impact performance and having a smaller packing depth might be beneficial. We also wanted to analyze its impact. Limiting the packing depth is required for NNLSHP to provide an appropriate bound for the complexity and make the algorithm tractable. SPFHP is not impacted by it as our experiments show and we provide a version that does not limit the packing depth. Since NNLSHP provided the best packing efficiency, we used it for our experiments to show how expected and measured speed-up match. There is no benefit in running the experiments additionally with the SPFHP packed data since it will be slower. Thus, we only provide a comparison of the two packing algorithms from the efficiency point of view (at reviewer EN5A).
- We agree with reviewer EN5A, that our explanation on learning rate scaling can be confusing. Hence, we replaced our reasoning by referring to our experiments that show that scaling the learning rate with the respective change in effective batch size reduces performance. This is in line with reference parametrizations for different batch sizes (https://github.com/mlcommons/logging/blob/master/mlperf_logging/rcp_checker/training_1.1.0/rcps_bert.json). If there are other references that recommend linear scaling, we would be eager to obtain references.
- Reviewer xs2k mentions the code complexity for packing which is a fair point. Note that no custom kernels are required in our approach which makes our approach much simpler than Efficient Transformer. Also note, that different frameworks already have some complexity due to “packing” code. Our approach could help to streamline approaches. The packed code can be always applied to unpacked data. This avoids the need for different approaches for pre-training and downstream tasks. To further address the code complexity, we are working on releasing respective code in different frameworks.
- It would be helpful if reviewer g5tJ could provide a clarification on point 6. As mentioned in the paper, our algorithm is hardware independent and can be transferred to GPU and TPU with different software frameworks like PyTorch and TensorFlow. In fact, we know that our approach has been applied in a different GPU application and for BERT it has been applied on new modern hardware beyond IPUs. Which additional details would be useful to provide in the paper?

References:
[1] Exploring the Limits of Transfer Learning with a Unified Text-to-Text Transformer

---

### Decision · Program_Chairs · 2022-01-20

**Decision:**

Reject

**Comment:**

This paper proposes to re-organize the training data in such a way that padding can be avoided. The novelty is somewhat limited and the results are what one would expect - a nice speed-up of 2x but nothing really game-changing. While the reviewer scores straddle the decision boundary, nobody is very strongly supportive of acceptance and the positive reviews actually have lower confidence.